# Estimation error of gradient descent in deep regressions

## Abstract

To achieve a theoretical understanding of deep learning, it is necessary to consider the approximation, generalization, and optimization errors. In recent years, there have been significant advancements in the literature regarding each or two of these errors. However, there have been few works that simultaneously analyze all three errors. This is due to the gap that exists between the optimization and generalization errors in over-parameterized regimes. In this work, we attempt to bridge this gap by establishing consistency between the outputs of gradient descent and the true regression function in the over-parameterized scenario. We provides the first error analysis that includes the approximation, generalization, and optimization errors in the scenario of deep regression. Our research offers a feasible perspective for a more comprehensive understanding of the theory behind deep learning.

## 1 Introduction

The success of deep learning in computer vision, natural language processing, and other fields has propelled the advancement of its theoretical research. It is now widely recognized that, as a non-parametric estimation method, error analysis in deep learning includes approximation error, generalization error, and optimization error Grohs & Kutyniok (2022); Telgarsky (2020); Weinan (2020); Bach (2021).

Approximation error refers to the difference between the target function class and the neural network function class used in the learning algorithm. The theoretical analysis of the approximation power of shallow networks dates back to the 1980s Cybenko (1989); Hornik et al. (1989); Hornik (1991), when people mainly focused on sigmoidal networks. In recent years, attention has shifted to ReLU networks due to their superior empirical performance in modern learning tasks. Yarotsky Yarotsky (2017) was the first to demonstrate how to construct a ReLU network that achieves any desired approximation accuracy using the Taylor expansion. Inspired by this, modern approximation results for deep neural networks have emerged, where network architecture parameters such as depth, width, and size are used to bound the approximation error Yarotsky (2017; 2018); Petersen & Voigtlaender (2018); Zhou (2020); Shen et al. (2019); Shen (2020); Lu et al. (2021); Gühring et al. (2020); Gühring & Raslan (2021); Siegel & Xu (2020). For more information, see Petersen (2020) and the references therein.

Classical methods in empirical process theory employ tools such as symmetrization and Lipschitz contraction to transform the study of generalization error into bounding the complexity of neural network classes, such as the Rademacher complexity, covering number, or VC-dimension. For detailed analysis, see Van Der Vaart et al. (1996); Van de Geer & van de Geer (2000); Giné & Nickl (2021). However, generalization analysis from the perspective of the uniform law of large numbers may lead to suboptimal error bounds Bartlett et al. (2017). Localized techniques that utilize the local structure of the hypothesis function class can reach sharp error bounds in scenarios where the Bernstein condition or off-set condition hold. For details, see Bartlett et al. (2005); Koltchinskii (2006); Mendelson (2018); Xu & Zeevi (2021); Kanade et al. (2022) and the references therein.

Recently, the use of deep neural networks for regression has garnered significant attention in the framework of nonparametric estimation Bauer & Kohler (2019); Kohler & Langer (2021); Imaizumi & Fukumizu (2019); Schmidt-Hieber (2020); Nakada & Imaizumi (2020); Farrell et al. (2021); Jiao et al. (2021); Suzuki (2019); Suzuki & Nitanda (2021); Shen et al. (2021); Fan et al. (2022). Optimal minimax estimation error has been derived in these works by adjusting the depth and width of the

neural network to balance approximation and generalization errors, building on the foundations of nonparametric estimation theory Stone (1982); Gyorfi et al. (2002); Tsybakov (2009). However, the convergence results in these elegant works only apply when the number of samples is larger than the number of parameters of the neural network.

Over-parameterized deep neural networks, where the number of parameters is much larger than the sample size, are widely used in real-world applications because they make model training computationally more efficient. Although the optimization problem in deep learning is highly non-convex, solvers such as (stochastic) gradient descent with randomized initialization and small step-size still converge linearly in modern over-parameterized regimes Jacot et al. (2018); Allen-Zhu et al. (2019); Du et al. (2019); Zou & Gu (2019); Liu et al. (2022); Chizat et al. (2019); Sun (2019); Zou et al. (2020); Oymak & Soltanolkotabi (2020); Nguyen & Mondelli (2020); Nguyen (2021) and the references therein. However, the reason why over-parameterized deep neural networks work well remains a mystery, and providing statistical guarantees in deep learning under over-parameterized regimes is still a theoretically fundamental but challenging problem Belkin (2021); Bartlett et al. (2021); Berner et al. (2021).

The current main perspective for understanding over-parameterization in linear and kernel models is the benign overfitting due to the double descent phenomenon for estimators that interpolate data with minimum norm Belkin et al. (2018; 2019a); Hastie et al. (2019); Belkin et al. (2019b); Liang & Rakhlin (2020); Nakkiran et al. (2020); Bartlett et al. (2020); Tsigler & Bartlett (2020); Belkin (2021); Bartlett et al. (2021); Belkin et al. (2018); Bartlett et al. (2020); Tsigler & Bartlett (2020). However, an interesting result was presented in Kohler & Krzyżak (2021), showing that the empirical risk minimization estimator (ERM) in nonparametric least squares regression with over-parameterized deep neural networks can be inconsistent when the distributions of covariates have no densities with respect to Lebesgue measure. Recently, the convergence rate of ERM with over-parameterization in deep regression has been derived for norm-controlled deep ReLU networks Jiao et al. (2023); Yang & Zhou (2023). However, the norm control in Jiao et al. (2023); Yang & Zhou (2023) for ERM is too restrictive, such that even randomized initialization will not satisfy the norm constraint. Therefore, there is still a gap to provide a comprehensive theoretical guarantee by considering all three errors (approximation, generalization, and optimization) simultaneously.

## 1.1 THE REGRESSION PROBLEM AND MAIN RESULT OF THIS PAPER

To the best of our knowledge, this paper provides the first error analysis that includes the approximation, generalization, and optimization errors in the scenario of deep regression. We first recall the problem setup. We have the regression model

$$Y = f^*(X) + \xi,$$

where $f^*(x) = \mathbb{E}[Y|X = x]$ is the unknown underlying target and $\xi$ is a sub-gaussian noise with $\mathbb{E}[\xi] = 0$ and $\|\xi\|_{\psi_2} \leq C_\xi$. At the papulation level, $f^*$ can be formulated as the solution of the least square problem

$$\min_f L(f) = \mathbb{E}_{(X,Y)}(f(X) - Y)^2. \tag{1}$$

In real data applications, the distribution of $(X, Y)$ is unknown, and we can only attempt to find an estimator based on a finite sample. Specifically, let $\{(X_i, Y_i)\}_{i=1}^n := \{Z_i\}_{i=1}^n$ be $n$ i.i.d. samples drawn from the unknown distribution. Let $m_0 := d, m_2 \in \mathbb{N}_{>0}$, we define a three-layer neural network class in the following form

$$\mathcal{F}_{NN}(\{B_3, B_2, B_1\}, m_2) := \left\{ \frac{1}{\sqrt{m_2}} \sum_{k_2=1}^{m_2} w_{k_2}^{(3)} \sigma\left( \sum_{k_1 \in I_{k_2}^{(2)}} w_{k_2 k_1}^{(2)} \sigma\left( \sum_{k_0=1}^{m_0} w_{k_1 k_0}^{(1)} x_{k_0} + b_{k_1}^{(1)} \right) + b_{k_2}^{(2)} \right) : \right.$$

$$\left. |w_{k_2}^{(3)}| \leq B_3, |w_{k_2 k_1}^{(2)}|, |b_{k_2}^{(2)}| \leq B_2, |w_{k_1 k_0}^{(1)}|, |b_{k_1}^{(1)}| \leq B_1 \right\}, \tag{2}$$

where the activate function is the logistic function $\sigma(x) = \frac{1}{1+e^{-x}}$,

$$I_{k_2}^{(2)} = \{(k_2 - 1)d + 1, (k_2 - 1)d + 2, \cdots, k_2 d\}.$$

The neuron numbers in the first, second and third layer are $m_0 = d$, $m_1 := m_2 d$ and $m_2$, respectively. Hence the width of neural networks in $\mathcal{F}_{NN}(\{B_3, B_2, B_1\}, m_2)$ is completely controlled by $m_2$. We use $W$ to denote the parameter vector (including weights and biases) of $f \in \mathcal{F}_{NN}(\{B_3, B_2, B_1\}, m_2)$. We define the empirical loss on $\mathcal{F}_{NN}(\{B_3, B_2, B_1\}, m_2)$

$$L_n(f_W) := \frac{1}{n} \sum_{i=1}^{n} (f_W(X_i) - Y_i)^2. \tag{3}$$

When we perform optimization algorithms over $\mathcal{F}_{NN}$, we treat the loss functional $L_n$ as a function of $W$, i.e., we define $L_n(W) := L_n(f_W)$ for any $f_W \in \mathcal{F}_{NN}$. In this paper, we focus on the global convergence of gradient descent. We believe our result can be extended to stochastic gradient descent. The iteration scheme of gradient descent under the regression model is

$$W_{t+1} = W_t - \eta \nabla_W L_n(W_t), \tag{4}$$

where $\eta$ is the learning rate, and $W_0$ is drawn from normal distributions according to

$$
\begin{aligned}
(w_0)_{k_2}^{(3)} &\sim N(0, \sigma_3^2), \quad k_2 \in [m_2]; \\
(w_0)_{k_2 k_1}^{(2)}, (b_0)_{k_2}^{(2)} &\sim N(0, \sigma_2^2), \quad k_1 \in I_{k_2}^{(2)}, k_2 \in [m_2]; \\
(w_0)_{k_1 k_0}^{(1)}, (b_0)_{k_1}^{(1)} &\sim N\left(0, \sigma_1^2\right), \quad k_0 \in [m_0], k_1 \in [m_1].
\end{aligned}
\tag{5}
$$

Define the kernel matrix

$$(K_0)_{ij} := \mathbb{E}_{(w,b) \sim \mathcal{N}\left(0, \sigma_1^2 I_{d+1}\right)}[\sigma\left(\langle w, X_i \rangle + b\right) \sigma\left(\langle w, X_j \rangle + b\right)], \quad i, j \in [n].$$

Let the truncating threshold $B_{tr} \in \mathbb{R}_{>0}$. We use

$$\hat{f}_t = f_{tr,B_{tr}}(f_{W_t}) \tag{6}$$

to denote the estimator at $t-$th step in gradient descent where the truncation function is defined as

$$
f_{tr,B_{tr}}(x) :=
\begin{cases}
-B_{tr}, & x \leq -B_{tr} \\
x, & -B_{tr} < x < B_{tr} \\
B_{tr}, & x \geq B_{tr}
\end{cases}.
$$

Our main result bounds the estimation error between the output of gradient descent in (6) and the target regression function $f^*$. We give an informal version first. The formal version of our main result is presented in Theorem 5.

**Theorem 1.** *(Informal) Assume $f^*$ is Lipschitz continuous. Let $n$ be sufficiently large. Set*

$$B_{tr} = \Theta(\log n), \quad m_2 = \mathcal{O}\left(n^{\frac{40d+40}{20d+19}}\right),$$

$$B_1 = \Theta(n^{2/(10d+9)}), \quad B_2 = \Theta(\log^2 n), \quad B_3 = \Theta(n^{3/2}),$$

$$t = \Omega(\log n), \quad L = \Theta(n^3 \log^4 n), \quad \mu = \Theta(1),$$

$$0 < \eta < \frac{2}{L}, \quad 0 < \eta\mu - \frac{1}{2}\eta^2\mu L < 1.$$

*Then with high probability,*

$$\|\hat{f}_t - f^*\|_{L^2(\nu)}^2 = \mathcal{O}(n^{-1/(10d+9)}).$$

## 1.2 CONTRIBUTION

In this work, we attempt to bridge the gap that exists between the optimization and generalization errors in over-parameterized regimes by establishing consistency between the outputs of gradient descent and the true regression function. We provides the first error analysis that includes the approximation, generalization, and optimization errors in the scenario of deep regression. Our research offers a feasible perspective for a more comprehensive understanding of the theory behind deep learning. To be specific, our findings establish a clear relationship with various parameters, including the dimension $d$, the variances in different layers $\sigma_1, \sigma_2, \sigma_3$, the truncation threshold $B_{tr}$, the bounds of samples $B_x, B_Y$, and the bound associated with the marginal distribution of samples

$B_\nu$. Moreover, our results offer explicit guidance on determining the appropriate width of the neural network, sample size, and learning rate. Additionally, we predict the estimated number of iterations required for gradient descent to attain a desired level of accuracy.

From a practical perspective, our results empower practitioners to optimize relevant factors based on the specific requirements of their task, ultimately improving the overall performance of the model. Furthermore, our results can guide researchers and practitioners in setting realistic expectations and planning their training processes more effectively. Overall, our results bridge the gap between theoretical analysis and practical application, furnishing valuable guidelines for optimizing deep neural networks in real-world scenarios.

### 1.3 ORGANIZATION

The organization of this paper is as follows: In section 2 we list our notations and assumptions used throughout the paper. Section 3 presents the main result along with a sketch of the proof. In Section 4, we summarize our findings and conclude the paper.

## 2 NOTATIONS AND ASSUMPTIONS

### 2.1 NOTATIONS

Let $\sigma$ be a general Lipschitz function. Define

$$L_\sigma := \max\{\text{Lipschitz constant of } \sigma, 1\}, \quad B_\sigma := \max\{\text{upper bound of } \sigma, 1\}.$$

The reason of making a comparison with 1 is to make the following theoretical results more concise. We define $L_{\sigma'}$ and $B_{\sigma'}$ in the same way. We define another index set

$$I_{k_1}^{(1)} = \left\{ \left\lceil \frac{k_1}{d} \right\rceil \right\}, k_1 \in [m_1].$$

Define $\widetilde{m}_1 := \left| I_{k_2}^{(2)} \right|, \widetilde{m}_2 := \left| I_{k_1}^{(1)} \right|$, then we immediately obtain $\widetilde{m}_1 = d, \widetilde{m}_2 = 1$. Denote the truncated version of $\mathcal{F}_{NN}(\{B_3, B_2, B_1\}, m_2)$ as

$$\mathcal{F}_{NN}^{(bounded)}(\{B_3, B_2, B_1\}, B_{tr}, m_2) := f_{tr, B_{tr}} \circ \mathcal{F}_{NN}(\{B_3, B_2, B_1\}, m_2).$$

To simplify notations, we write $\mathcal{F}_{NN}^{(bounded)}(\{B_3, B_2, B_1\}, B_{tr}, m_2)$ as $\mathcal{F}_{NN}^{(bounded)}$ and $\mathcal{F}_{NN}(\{B_3, B_2, B_1\}, m_2)$ as $\mathcal{F}_{NN}$ below when there is no ambiguity.

### 2.2 ASSUMPTIONS

Denote $B_x := \max\{\max_{i\in[n],j\in[d]} |(X_i)_j|, 1\}$. Let $B_\mathcal{F}, L_\mathcal{F} \in \mathbb{R}_{\geq 1}$. Define

$$\mathcal{F} := \{f : [-B_x, B_x]^d \to \mathbb{R} \mid f \text{ is } L_\mathcal{F}\text{-Lipschitz continuous and } |f| \leq B_\mathcal{F}\}.$$

We make the following assumption to the target function $f^*$.

**Assumption 1.** *The target $f^*$ lies in $\mathcal{F}$.*

Under assumption 1 and $Y_i = f^*(X_i) + \xi_i$ we know that $Y_i$ is a sub-gaussian random variable with $\|Y_i\|_{\psi_2} \leq C \left( \frac{1}{\ln 2} B_\mathcal{F}^2 + C_\xi^2 \right)^{1/2}$. By Proposition 2.5.2 in Vershynin (2018) and union bound we have with probability at least $1 - t$,

$$B_Y := \max \left\{ \max_{i\in[n]} |Y_i|, 1 \right\} \leq C \left( \frac{1}{\ln 2} B_\mathcal{F}^2 + C_\xi^2 \right)^{1/2} \ln^{1/2} \frac{2n}{t}.$$

Let $\nu$ be the marginal distribution of $X$ and $f_\nu$ the corresponding density function.

**Assumption 2.** *$f_\nu$ is $L^2$-integrable with respect to Lebesgue measure, i.e., there exists $B_\nu < \infty$ such that*

$$\int_{[-B_x, B_x]^d} f_\nu^2(x) dx \leq B_\nu.$$

**Remark 1.** *The two assumptions are weak. We only assume the target lies in Lipschitz continuous and the density function of the samples is $L_2$ Lebesgue integrable. Comparing with some other works that assume the target lies in Barron space or some other spaces with more structures, our Lipschitz assumption is relatively weak. As a result, our results have broader applicability and generality. In fact, the Lipschitz assumption is a standard assumption in nonparametric estimation, for example, see Gyorfi et al. (2002).*

## 3 ERROR DECOMPOSITION, PROOF SKETCH AND MAIN RESULTS

### 3.1 ERROR DECOMPOSITION

The following proposition decomposes the total estimation error $\|\hat{f}_t - f^*\|^2_{L^2(\nu)}$, i.e., the error between the output of gradient descent in (6) and the target regression function, into the approximation, generalization, and optimization errors.

**Proposition 1.** *Let $\nu$ be the marginal distribution of $X$. We have*

$$\|\hat{f}_t - f^*\|^2_{L^2(\nu)} \le$$

$$\underbrace{\inf_{f \in \mathcal{F}_{NN}^{(bounded)}} \|f - f^*\|^2_{L^2(\nu)}}_{\mathcal{E}_{app}} + \underbrace{\sup_{f \in \mathcal{F}_{NN}^{(bounded)}} [L(f) - L_n(f)] + \sup_{f \in \mathcal{F}_{NN}^{(bounded)}} [L_n(f) - L(f)]}_{\mathcal{E}_{gen}} + \underbrace{L_n(\hat{f}_t)}_{\mathcal{E}_{opt}}.$$

**Remark 2.** *The first term shows the distance between the set $\mathcal{F}_{NN}$ and the target function $f^*$, which is exactly the classical definition of approximation error. The second term is the generalization error, which measures the uniform difference between the population loss in (1) and the empirical loss in (3). The last term reflects how small the empirical loss value at the $t$-th iteration can be, and is therefore called the optimization error.*

### 3.2 PROOF SKETCH

Based on the above Proposition 1, we need to bound the approximation error, generalization error and optimization error, respectively. Here we present some significant intermediate results. The proof details are all postponed to the supplementary material.

We first define some notations to shorten the presentation. Set $B_{tr} \ge \max\{B_Y, B_{\mathcal{F}} + 3\}$. Let $t_1, t_2, t_3, t_4, t_5, t_6 > 0, 0 < \epsilon < 2^{d/4+1} 3^{1/2} B_x^{d/4} B_\nu^{1/2}$. Let

$$\alpha = \max\left\{ \frac{2^{d/4+2} 3^{1/2} B_x^{d/4} B_\nu^{1/2} B_{\mathcal{F}}}{\epsilon} \left( \left\lceil \frac{2^{d/4+1} 3^{1/2} d^{1/2} B_x^{d/4} B_\nu^{1/2} L_{\mathcal{F}}}{\epsilon} \right\rceil \right)^d - 1, e^{2d+1} \right\},$$

$$\delta = \frac{15\epsilon^4}{2d \left( \epsilon + 2^{d/4+2} 3^{1/2} B_x^{d/4} B_\nu^{1/2} \right)^4} \frac{1}{\left\lceil \frac{2^{d/4+1} 3^{1/2} d^{1/2} B_x^{d/4} B_\nu^{1/2} L_{\mathcal{F}}}{\epsilon} \right\rceil}.$$

Denote

$$B_1^{(t_6)} = 2C^{1/2} \sigma_1 \ln^{1/2} \frac{d^2 m_2}{t_6}, \qquad \bar{B}_1 = \max\left\{ \frac{1}{B_x}, \delta + 3 \right\} \frac{\ln\ln\alpha}{\delta},$$

$$B_2^{(t_3)} = 2C^{1/2} \sigma_2 \ln^{1/2} \frac{d m_2}{t_3}, \qquad \bar{B}_2 = (2d+2) \ln^2 \alpha + (2d-1) \ln \alpha,$$

$$B_3^{(t_2)} = 2C^{1/2} \sigma_3 \ln^{1/2} \frac{m_2}{t_2}, \qquad \bar{B}_3 = \sqrt{m_2} B_{\mathcal{F}}$$

and

$$B_1 = \max\left\{ B_1^{(t_6)}, \bar{B}_1 \right\}, \quad B_2 = \max\left\{ B_2^{(t_3)}, \bar{B}_2 \right\}, \quad B_3 = \max\left\{ B_3^{(t_2)}, \bar{B}_3 \right\}.$$

Let

$$C_1 := 2^{1/4} (3B_{tr} + 2B_Y)^{1/4} m_2^{1/8} B_x^d B_3^{1/4} B_2^d B_1^d \left( \max_{s_1 \in \{1,2,\cdots,4d\}} B_{\sigma^{(s_1)}} \right)^{\frac{1}{4}(4d+1)} \mathcal{B}_{4d}^{1/2}$$

$$\left(\frac{5d-1}{d-1}\right)^{1/4}\left(\frac{5d-2}{d-1}\right),$$

$$C_2 := \frac{2enB_{tr}(3B_{tr}+2B_Y)}{C_1},$$

$$C_3 := 2\int_0^\infty \left[1 - 2e^{-t^2/(C(d+1)\sigma_2^2)}\right]\sigma'(t)\sigma''(t)dt.$$

where $\mathcal{B}_j$ is the Bell number. Set

$$\mu = C_3\widetilde{m}_1\sigma_3^2\lambda_{\min}(K_0)$$

and

$$L = 32d^2\left[\sqrt{\frac{86}{m_2}}d\left(\frac{2\sigma_3^2 B_\sigma^2}{C}\ln\frac{2}{t_5}+2B_Y^2\right)^{1/2}+\sqrt{5(d^2+2d+2)}\right]\left(B_3^{(t_2)}\right)^2\left(B_2^{(t_3)}\right)^2.$$

Denote

$$n_1^{(low)} = \left(\frac{192}{7}\right)^2\frac{144C_1}{\epsilon^4}\left(\frac{(B_{tr}+B_Y)^2}{2}\right)^{7/4}\left(\ln\frac{2^{3/4}C_2}{(B_{tr}+B_Y)^{3/2}}+\frac{6}{7}\right)^2,$$

$$n_2^{(low)} = \frac{144(B_{tr}+B_Y)^4}{\epsilon^4}\ln\frac{2}{t_1},$$

$$n_3^{(low)} = \frac{C_1}{2B_{tr}^{1/4}(3B_{tr}+2B_Y)^{1/4}},$$

$$n_4^{(low)} = C_1\max\left\{\left(\frac{14}{\ln\frac{2^{3/4}C_2}{(B_{tr}+B_Y)^{3/2}}+\frac{6}{7}}\right)^{2/7},1\right\}\left(\frac{2}{(B_{tr}+B_Y)^2}\right)^{1/4}$$

and

$$m_{2,1}^{(low)} = \frac{n\left(B_3^{(t_2)}\right)^2\log_{e/2}\frac{n}{t_4}}{2C_3d^5\sigma_3^2\lambda_{\min}(K_0)},$$

$$m_{2,2}^{(low)} = \frac{344(d^2+2d+2)n^2B_x^6\left(\max\left\{B_3^{(t_2)},B_2^{(t_3)}\right\}\right)^{12}}{C_3^2d^2\sigma_3^4\lambda_{\min}^2(K_0)},$$

$$m_{2,3}^{(low)} = \frac{64d\left(B_3^{(t_2)}\right)^2\left(B_2^{(t_3)}\right)^2\left(\frac{2\sigma_3^2}{C}\ln\frac{2}{t_5}+2B_Y^2\right)}{(\mu-\frac{1}{2}\eta\mu L)^2\left(\min\left\{B_1^{(t_6)},B_2^{(t_3)},B_3^{(t_2)}\right\}\right)^2}.$$

We first construct a third layer network contained in $\mathcal{F}_{NN}(\{B_3,B_2,B_1\},m_2)$ that can approximate the Lipschitz target arbitrarily well.

**Theorem 2.** *Let Assumption 1 holds. For any $f \in \mathcal{F}$, there exists $g \in \mathcal{F}_{NN}(\{\bar{B}_3,\bar{B}_2,\bar{B}_1\},m_2)$ such that*

$$\|f-g\|_4 \le \epsilon,$$

$$\|f-g\|_\infty \le 2 + \frac{\epsilon}{2^{d/4+1}B_x^{d/4}}.$$

For the generalization error we first drive an upper bound in expectation using symmetrization, Lipschitz contraction and chaining. Then, we get a high probability bound via McDiarmid's inequality.

**Theorem 3.** *Let Assumption 1,2 hold. Let $n \ge \max\left\{n_1^{(low)},n_2^{(low)},n_3^{(low)}\right\}$. Then with probability at least $1 - t_1$ over $\{Z_i\}_{i=1}^n$,*

$$\mathcal{E}_{gen} \le \frac{384}{7}\left(\frac{C_1}{n}\right)^{1/2}\left(\frac{(B_{tr}+B_Y)^2}{2}\right)^{7/8}\left(\ln\frac{2^{3/4}C_2}{(B_{tr}+B_Y)^{3/2}}+\frac{6}{7}\right)+\frac{\epsilon^2}{12}.$$

Note that the existing theory for optimization in deep learning focuses on unconstrained problems Jacot et al. (2018); Allen-Zhu et al. (2019); Du et al. (2019); Zou & Gu (2019); Liu et al. (2022); Chizat et al. (2019); Sun (2019). However, the optimization error considered here is a constrained problem, which requires more effort to handle. We first derive a PL-inequality around the initialization, as long as the network is wide enough. Then, we show that the iterates $W_t$ fall into a neighborhood of the initialization if we set the step-size and constraint parameters $B_1, B_2, B_3$ properly. Finally, we can derive the following linear convergence result.

**Theorem 4.** *Initialize $W_0$ by (5). Let Let $\eta$ satisfy $0 < \eta < \frac{2}{L}$ and $\eta\mu - \frac{1}{2}\eta^2\mu L < 1$. If*

$$m_2 \geq \max\left\{m_{2,1}^{(low)}, m_{2,2}^{(low)}, m_{2,3}^{(low)}\right\},$$

*then for any $t \in \mathbb{N}_{\geq 0}$, there holds*

$$|w_t - w_0| \leq \min\left\{B_1^{(t_6)}, B_2^{(t_3)}, B_3^{(t_2)}\right\},$$

$$L_n(W_t) \leq \left(1 - \eta\mu + \frac{1}{2}\eta^2\mu L\right)^t \left(\frac{2\sigma_3^2 B_\sigma^2}{C}\ln\frac{2}{t_5} + 2B_Y^2\right)$$

*with probability at least $1 - t_2 - t_3 - t_4 - t_5 - t_6$ over $W_0$.*

## 3.3 MAIN RESULTS

Combining Theorem 2-4 we get our main results.

**Theorem 5.** *Let Assumption 1,2 hold. Initialize $W_0$ by (5). Choose the learning rate $\eta$ satisfy*

$$0 < \eta < \frac{2}{L}, \quad 0 < \eta\mu - \frac{1}{2}\eta^2\mu L < 1,$$

*then we have with probability at least $1 - t_1$ over $\{Z_i\}_{i=1}^n$ and $1 - t_2 - t_3 - t_4 - t_5 - t_6$ over $W_0$,*

$$\|f_{tr,B_{tr}} \circ f_{W_t} - f^*\|_{L^2(\nu)} \leq \epsilon$$

*as long as*

$$t \geq \log_2\frac{6\left(\frac{2\sigma_3^2}{C}\ln\frac{2}{t_5} + 2B_Y^2\right)}{\epsilon^2},$$

*and the sample size satisfies*

$$n \geq \max\left\{n_1^{(low)}, n_2^{(low)}, n_3^{(low)}, n_4^{(low)}\right\}$$

*and the width satisfies*

$$m_2 \geq \max\left\{m_{2,1}^{(low)}, m_{2,2}^{(low)}, m_{2,3}^{(low)}\right\}.$$

*Proof.* Proposition 1 decomposes the total error into three kinds of errors. We prove that under our setting of parameters, each error is bounded by $\frac{\epsilon^2}{3}$ by using the above results.

We first take a look at the approximation error. We know from Theorem 2 that there exists $g \in \mathcal{F}_{NN}(\{\bar{B}_3, \bar{B}_2, \bar{B}_1\}, m_2)$ such that $\|f^* - g\|_4 \leq \frac{\epsilon}{\sqrt{3B_\nu}}$ and $\|f^* - g\|_\infty \leq 2 + \frac{\epsilon}{2^{d/4+1}3^{1/2}B_\nu^{1/2}B_x^{d/4}}$. In fact, $g$ also lies in $\mathcal{F}_{NN}^{(bounded)}(\{B_3, B_2, B_1\}, B_{tr}, m_2)$ because for any $x \in [-B_x, B_x]^d$,

$$|g(x)| \leq |f^*(x) - g(x)| + |f^*(x)| \leq 2 + \frac{\epsilon}{2^{d/4+1}3^{1/2}B_\nu^{1/2}B_x^{d/4}} + B_\mathcal{F} \leq 3 + B_\mathcal{F} \leq B_{tr}$$

provided $\epsilon \leq 2^{d/4+1}3^{1/2}B_x^{d/4}B_\nu^{1/2}$. By Hölder inequality and Assumption 2 we can bound the approximation error which appears in the form of $L^2(\nu)$-norm:

$$\|f^* - g\|_{L^2(\nu)}^2 = \int_{[-B_x, B_x]^d} |f^*(x) - g(x)|^2 f_\nu(x)dx$$

$$\leq \left( \int_{[-B_x, B_x]^d} |f^*(x) - g(x)|^4 dx \right)^{1/2} \left( \int_{[-B_x, B_x]^d} f_\nu^2(x) dx \right)^{1/2}$$

$$\leq B_\nu \|f^* - g\|_4^2 \leq B_\nu \cdot \frac{\epsilon^2}{3B_\nu} \leq \frac{\epsilon^2}{3}. \tag{7}$$

Next, we check the optimization error. Let $\eta$ satisfies $0 < \eta < \frac{2}{L}$ and $0 < \rho = \eta\mu - \frac{1}{2}\eta^2\mu L < 1$ in Theorem 4, then by our choice of $m_2$ and $t$, we have with probability at least $1 - t_2 - t_3 - t_4 - t_5$ over $W_0$,

$$L_n(W_t) \leq (1 - \rho)^t \left( \frac{2\sigma_3^2}{C} \ln \frac{2}{t_5} + 2B_Y^2 \right) \leq \frac{\epsilon^2}{3}.$$

The truncation operation further decreases the loss value. Let's clarify this fact. For $i \in [n]$, if $|f_W(X_i)| \leq B_{tr}$, then $|f_{tr,B_{tr}} \circ f_W(X_i) - Y_i| = |f_W(X_i) - Y_i|$; if $|f_W(X_i)| > B_{tr}$, without loss of generality we assume $f_W(X_i) > B_{tr}$. Since $Y_i \leq B_Y \leq B_{tr}$,

$$|f_W(X_i) - Y_i| = |[f_W(X_i) - B_{tr}] + [B_{tr} - Y_i]| = [f_W(X_i) - B_{tr}] + [B_{tr} - Y_i]$$
$$> B_{tr} - Y_i = |B_{tr} - Y_i| = |f_{tr,B_{tr}} \circ f_W(X_i) - Y_i|.$$

Hence with the same probability,

$$L_n(f_{tr,B_{tr}} \circ f_{W_t}) \leq L_n(f_{W_t}) = L_n(W_t) \leq \frac{\epsilon^2}{3}. \tag{8}$$

An important thing is to make sure that $f_{W_t}$ lies in $\mathcal{F}_{NN}(\{B_3, B_2, B_1\}, m_2)$. In fact, Theorem 4 tells us that with the same probability,

$$B_3(W_t) = \max \left\{ \max_{k_2} \left| (w_t)_{k_2}^{(3)} \right|, 1 \right\}$$

$$\leq \max \left\{ \max_{k_2} \left| (w_t)_{k_2}^{(3)} - (w_0)_{k_2}^{(3)} \right| + \left| (w_0)_{k_2}^{(3)} \right|, 1 \right\} \leq \max \left\{ B_3^{(t_6)}, 1 \right\} \leq B_3,$$

and similarly $B_2(W_t) \leq B_2, B_1(W_t) \leq B_1$. Hence $f_{W_t} \in \mathcal{F}_{NN}(\{B_3, B_2, B_1\}, m_2)$ with the same probability.

Finally, we can immediately obtain an upper bound of generalization error from Theorem 3. In this case, by our choice of $n$ we have with probability at least $1 - t_1$ over $\{Z_i\}_{i=1}^n$,

$$\sup_{f \in \mathcal{F}_{NN}^{(bounded)}} \left[ \frac{1}{n} \sum_{i=1}^n [f(X_i) - Y_i]^2 - \mathbb{E}[f(X) - Y]^2 \right] \leq \frac{\epsilon^2}{6}, \tag{9}$$

$$\sup_{f \in \mathcal{F}_{NN}^{(bounded)}} \left[ \mathbb{E}[f(X) - Y]^2 - \frac{1}{n} \sum_{i=1}^n [f(X_i) - Y_i]^2 \right] \leq \frac{\epsilon^2}{6}. \tag{10}$$

We finish the proof by combining Proposition 1 and (7)-(10). $\qquad \square$

## 4 CONCLUSION AND FUTURE WORK

In this paper, we attempt to bridge the gap that exists between the optimization and generalization errors in over-parameterized regimes. We establish the first consistency between the outputs of gradient descent and the true regression function in the over-parameterized scenario. Our research offers a feasible perspective for a more comprehensive understanding of the theory behind deep learning.

We hope that this work can motivate more studies on statistical theories of over-parametrized deep learning. In the following, we list several issues for future research.

First, in this work, we focus on gradient descent in regression for a three-layer network class. In practical application scenarios, researchers predominantly utilize SGD as the optimizer. How to generalize the current results to the case of SGD is an intriguing question. Currently, there is still

a gap in theory regarding how to transition from GD to SGD. The transition from GD to SGD introduces additional complexities and considerations that need to be addressed. While GD optimizes the parameters using the full dataset in each iteration, SGD randomly samples a subset of data, leading to more efficient computation but potentially introducing additional noise and variance in the optimization process. The convergence properties, generalization performance, and optimization dynamics of SGD differ from GD, making it necessary to bridge the theoretical gap in understanding this transition. In fact, this is exactly the problem we are currently working on solving. To address this gap, further research is needed to investigate the convergence behavior and optimization guarantees of SGD in the context of regression for three-layer networks. Exploring the theoretical foundations and developing analytical frameworks that capture the unique characteristics of SGD will be crucial. Additionally, empirical studies and experiments can provide insights into the practical performance of SGD in real-world scenarios and help validate and refine theoretical findings.

Second, the estimation error rate in this paper, $\mathcal{O}(n^{-1/(10d+9)})$, highlights a significant achievement in estimating the Lipschitz target. However, it is essential to acknowledge that there exists a gap between this rate and the minimax optimal rate of $\mathcal{O}(n^{-2/(d+2)})$. This difference indicates that further efforts are required to enhance the analysis presented in this work. Closing this gap and achieving the minimax optimal rate would not only consolidate the theoretical foundations but also contribute to the practical applicability of the proposed estimation techniques. Future research endeavors should focus on exploring innovative methodologies and refining the analysis to bridge this gap and approach the optimal estimation rate. By addressing this discrepancy, we can advance the field and provide more accurate and efficient estimation methods for Lipschitz targets.

Third, this study focuses on three-layer networks, and there exists a disparity between the theoretical findings and their practical application, particularly considering the prevalent use of deep neural networks in various learning tasks. Therefore, extending the current results to deep neural networks is of paramount importance, and it is precisely the aspect we are currently dedicated to investigating.

Fourth, this study leverages the smoothness of the sigmoidal networks to derive a consistent error bound. On the other hand, in modern learning tasks, although sigmoidal networks continue to play a vital role, the empirical performance of ReLU networks has attracted significant attention. It is challenging to establish such a bound for ReLU networks due to the lack of sufficient smoothness. While consistency convergence results may still exist for ReLU networks, they require different analytical techniques which is worth investing effort in studying.

Fifth, the current work primarily focuses on regression problems. However, it is important to note that there are numerous other types of learning tasks that warrant investigation. For instance, classification tasks, as well as solving partial differential equations (PDEs), are areas that require further study. Understanding how the findings of this research can be extended and applied to these different learning tasks is an important and intriguing direction for future exploration. Exploring the applicability of the current results in various learning domains will contribute to a more comprehensive understanding of the broader impact and potential of the proposed techniques.

Finally, conducting empirical studies to support the theoretical analysis is essential for validating and strengthening the findings of our work. We acknowledge the importance of empirical validation in evaluating the practical performance of the proposed methods. To address this limitation, we will incorporate empirical studies in our future research. By conducting experiments and evaluating the performance of the proposed techniques on real-world datasets, we can provide empirical evidence to complement and reinforce the theoretical analysis. This empirical validation will enhance the credibility and applicability of our work, ensuring that the proposed methods are not only theoretically sound but also effective in practical scenarios.

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
