# SUPPLEMENTARY MATERIAL: ESTIMATION ERROR OF GRADIENT DESCENT IN DEEP REGRESSIONS

In this supplementary material, we provide a detailed proof of Theorem 2-4. Before we begin, let us recall some background in real analysis and high-dimensional probability.

## A PRELIMINARIES

The function spaces we need to deal with are the $L^p$ spaces.

**Definition A.1** ($L^p$ space). *The $L^p$ spaces and corresponding norms are defined as follows*

$$L^p(\Omega) := \left\{ f : \Omega \to \mathbb{R} \mid \int_\Omega f^p dx < \infty \right\}, \quad \|f\|_p := \left[ \int_\Omega f^p(x) dx \right]^{1/p}, \quad p \in [1, \infty),$$

$$L^\infty(\Omega) := \{ f : \Omega \to \mathbb{R} \mid \exists C > 0 \ s.t. \ |f| \le C \ a.e. \}, \quad \|f\|_\infty := \inf\{ C \ge 0 \mid |f| \le C \ a.e. \}.$$

**Lemma A.1** (Hölder inequality). *Assume $p, q \in \mathbb{R}_{>0}$ and $1/p + 1/q = 1$. Let $f \in L^p(\Omega)$ and $f \in L^q(\Omega)$. Then $fg \in L^1(\Omega)$ and*

$$\|fg\|_1 \le \|f\|_p \|g\|_q.$$

We then introduce some concepts in high-dimension probability theory.

**Definition A.2** (sub-gaussian random variables). *A random variable $X$ is called sub-gaussian if there exists constant $C$ such that*

$$\mathbb{E} e^{X^2/C^2} \le 2.$$

*The sub-gaussian norm of $X$, denoted as $\|X\|_{\psi_2}$, is defined to be the smallest $C$:*

$$\|X\|_{\psi_2} := \inf \left\{ C > 0 : \mathbb{E} e^{X^2/C^2} \le 2 \right\}.$$

**Definition A.3** (sub-gaussian random variables in higher dimensions). *A random vector $X$ in $\mathbb{R}^d$ is called sub-gaussian if the one-dimensional marginals $\langle X, x \rangle$ are sub-gaussian random variables for all $x \in \mathbb{R}^d$. The sub-gaussian norm of $X$ is defined as $\|X\|_{\psi_2} := \sup_{x \in S^{d-1}} \|\langle X, x \rangle\|_{\psi_2}$, where $S^{d-1}$ is Euclidean unit sphere in $\mathbb{R}^d$ with center at the origin.*

**Lemma A.2** (McDiarmid's inequality). *Let $g$ be a function from $\Omega^n$ to $\mathbb{R}$. Suppose that function $g$ satisfies the bounded differences property: there exists constants $\{c_i\}_{i=1}^n$ such that for any $x_1, \cdots, x_n \in \Omega$,*

$$\sup_{\widetilde{x}_i \in \Omega} |g(x_1, \cdots, \widetilde{x}_i, \cdots, x_n) - g(x_1, \cdots, x_i, \cdots, x_n)| \le c_i, \quad i \in [n].$$

*Let $\{X_i\}_{i=1}^n$ be independent variables, then for any $\tau > 0$,*

$$|g(X_1, \cdots, X_n) - \mathbb{E} g(X_1, \cdots, X_n)| \le \tau.$$

*with probability at least $1 - 2e^{-\frac{2\tau^2}{\Sigma_{i=1}^n c_i^2}}$.*

The following theorem is useful for estimating the eigenvalues of a matrix.

**Lemma A.3** (Weyl's inequality). *Let $M = N + R$ with $M, N, R$ being $\mathbb{R}^{n_1 \times n_1}$ Hermitian matrices. Denote $\{\lambda_i(M)\}_{i=1}^{n_1}, \{\lambda_i(N)\}_{i=1}^{n_1}, \{\lambda_i(R)\}_{i=1}^{n_1}$ as the eigenvalues of $M, N, R$ with a decreasing order, respectively. Then the following inequalities hold:*

$$\lambda_i(N) + \lambda_n(R) \le \lambda_i(M) \le \lambda_i(N) + \lambda_1(R), \quad i \in [n_1].$$

Next, we introduce some concepts in function complexity theory.

**Definition A.4** (Rademacher complexity). *The Rademacher complexity of a set $A \subseteq \mathbb{R}^n$ is defined as*

$$\mathfrak{R}_n(A) = \mathbb{E}_{\{\varsigma_i\}_{i=1}^n} \left[ \sup_{a \in A} \frac{1}{N} \sum_{i=1}^n \varsigma_i a_i \right],$$

*where, $\{\varsigma_i\}_{i=1}^n$ are $n$ i.i.d Rademacher variables with $\mathbb{P}(\varsigma_i = 1) = \mathbb{P}(\varsigma_i = -1) = \frac{1}{2}$. The Rademacher complexity of function class $\mathcal{G}$ associate with random sample $\{X_i\}_{i=1}^n$ is defined as*

$$\mathfrak{R}_n(\mathcal{G}) = \mathbb{E}_{\{X_i, \varsigma_i\}_{i=1}^n} \left[ \sup_{u \in \mathcal{G}} \frac{1}{n} \sum_{i=1}^n \varsigma_i u(X_i) \right].$$

**Definition A.5** (covering number). *An $\epsilon$-cover of a set $T$ in a metric space $(S, \tau)$ is a subset $T_c \subset S$ such that for each $t \in T$, there exists a $t_c \in T_c$ such that $\tau(t, t_c) \leq \epsilon$. The $\epsilon$-covering number of $T$, denoted as $\mathcal{N}(\epsilon, T, \tau)$ is defined to be the minimum cardinality among all $\epsilon$-cover of $T$ with respect to the metric $\tau$.*

**Definition A.6** (uniform covering number). *Let $\mathcal{G}$ be a function class. Given $n$ sample $\{X_i\}_{i=1}^n$, define $\mathcal{G}|_{\{X_i\}_{i=1}^n} = \{(g(X_1), g(X_2), \cdots, g(X_n)) : g \in \mathcal{G}\}$. The uniform covering number $\mathcal{N}(\epsilon, \mathcal{G}, \| \cdot \|_\infty, n)$ is then defined by*

$$\mathcal{N}(\epsilon, \mathcal{G}, \| \cdot \|_\infty, n) := \max_{\{X_i\}_{i=1}^n} \mathcal{N}(\epsilon, \mathcal{G}|_{\{X_i\}_{i=1}^n}, \| \cdot \|_\infty).$$

**Definition A.7** (pseudo-dimension). *Let $\mathcal{G}$ be a class of functions from $\Omega$ to $\mathbb{R}$. Suppose that $S = \{x_1, x_2, \cdots, x_n\} \subset \Omega$. We say that $S$ is pseudo-shattered by $\mathcal{G}$ if there exists $y_1, y_2, \cdots, y_n$ such that for any $b \in \{0, 1\}^n$, there exists a $g \in \mathcal{G}$ satisfying*

$$\text{sign}(g(x_i) - y_i) = b_i, \quad i = 1, 2, \dots, n$$

*and we say that $\{y_i\}_{i=1}^n$ witnesses the shattering. The pseudo-dimension of $\mathcal{G}$, denoted as $\text{Pdim}(\mathcal{G})$, is defined to be the maximum cardinality among all sets pseudo-shattered by $\mathcal{G}$.*

## B  PROOF OF PROPOSITION 1: ERROR DECOMPOSITION

*Proof of Proposition 1.* Let $f$ be any function in $\mathcal{F}_{NN}^{(bounded)}$.

$$
\begin{aligned}
L(\hat{f}_t) =& [L(\hat{f}_t) - L_n(\hat{f}_t)] + [L_n(\hat{f}_t) - L_n(f)] + [L_n(f) - L(f)] + L(f) \\
\leq& [L(\hat{f}_t) - L_n(\hat{f}_t)] + L_n(\hat{f}_t) + [L_n(f) - L(f)] + L(f) \\
\leq& L(f) + \sup_{f \in \mathcal{F}_{NN}^{(bounded)}} [L(f) - L_n(f)] + \sup_{f \in \mathcal{F}_{NN}^{(bounded)}} [L_n(f) - L(f)] + L_n(\hat{f}_t).
\end{aligned}
$$

Note that

$$
\begin{aligned}
L(f) &= \mathbb{E}_{X,Y}(f(X) - Y)^2 = \mathbb{E}_{X,\xi}(f(X) - f^*(X) - \xi)^2 \\
&= \mathbb{E}_X(f(X) - f^*(X))^2 - 2\mathbb{E}_{X,\xi}(f(X) - f^*(X))\xi + \mathbb{E}_\xi \xi^2 \\
&= \mathbb{E}_X(f(X) - f^*(X))^2 + V^2.
\end{aligned}
$$

Hence

$$
\begin{aligned}
&\|\hat{f}_t - f^*\|_{L^2(\nu)}^2 \\
&\leq \|f - f^*\|_{L^2(\nu)}^2 + \sup_{f \in \mathcal{F}_{NN}^{(bounded)}} [L(f) - L_n(f)] + \sup_{f \in \mathcal{F}_{NN}^{(bounded)}} [L_n(f) - L(f)] + L_n(\hat{f}_t).
\end{aligned}
$$

We complete the proof by taking the infimum of the first term on the right-hand side over $\mathcal{F}_{NN}^{(bounded)}$. $\square$

## C PROOF OF THEOREM 2: APPROXIMATION POWER OF NEURAL NETWORK CLASSES

In this section, we study the approximation power of neural networks with the activation being logistic function:

$$\sigma(x) = \frac{1}{1 + e^{-x}}.$$

**Remark 1.** *The approximation results of neural networks activated by other sigmoidal type activations(such as $\tanh(x), \arctan(\text{x})$) can be established in similar ways.*

We first establish an approximation result on $[0, 1]^d$ and then derive the result on $[-B_x, B_x]^d$.

Define

$$
\begin{aligned}
g_{\bar{k}}^{(1)}(x) =& \sigma\left(\frac{2\ln\ln\alpha}{\delta}x - \frac{2\bar{k}\ln\ln\alpha}{N\delta} - \ln\ln\alpha\right) \\
&+ \sigma\left(-\frac{2\ln\ln\alpha}{\delta}x + \frac{2(\bar{k}+1)\ln\ln\alpha}{N\delta} - \ln\ln\alpha\right) - 1, \quad \bar{k} = 0, 1, \cdots, N-1 \quad \text{(C.1)}
\end{aligned}
$$

where parameters $N, \delta, \alpha$ will be defined later.

**Lemma C.1.** *Let $\bar{k} = 0, 1, \cdots, N-1$.*

$$
g_{\bar{k}}^{(1)}(x) \in \left(\sigma\left(\frac{2\ln\ln\alpha}{N\delta} - \ln\ln\alpha\right) - 1, \frac{1}{1+\ln\alpha}\right), \qquad x < \frac{\bar{k}}{N} \ or \ x > \frac{\bar{k}+1}{N}
$$

$$
g_{\bar{k}}^{(1)}(x) \in \left(1 - \frac{2}{1+\ln n}, 2\sigma\left(\frac{2\ln\ln\alpha}{N\delta} - 3\ln\ln\alpha\right) - 1\right), \qquad \frac{\bar{k}}{N} + \delta < x < \frac{\bar{k}+1}{N} - \delta
$$

*Proof.* Let

$$
h_1 = \sigma\left(\frac{2\ln\ln\alpha}{\delta}x - \frac{2\bar{k}\ln\ln\alpha}{N\delta} - \ln\ln\alpha\right), h_2 = \sigma\left(-\frac{2\ln\ln\alpha}{\delta}x + \frac{2(\bar{k}+1)\ln\ln\alpha}{N\delta} - \ln\ln\alpha\right).
$$

Then $g_{\bar{k}}^{(1)} = h_1 + h_2 - 1$. When $x < \frac{\bar{k}}{N}$,

$$
0 < h_1 < \sigma(-\ln\ln\alpha) = \frac{1}{1+\ln\alpha}, \quad \sigma\left(\frac{2\ln\ln\alpha}{N\delta} - \ln\ln\alpha\right) < h_2 < 1.
$$

When $x > \frac{\bar{k}+1}{N}$, it is a symmetric case. When $\frac{\bar{k}}{N} + \delta < x < \frac{\bar{k}+1}{N} - \delta$,

$$
1 - \frac{1}{1+\ln\alpha} = \sigma(\ln\ln\alpha) < h_1, h_2 < \sigma\left(\frac{2\ln\ln\alpha}{N\delta} - 3\ln\ln\alpha\right).
$$

$\square$

Let $k = (k_1, \cdots, k_d)$ with $k_i = 0, 1, \cdots, N-1 (i \in [d])$. Define

$$
\Omega_k^{out} = \left(\frac{k_1}{N}, \frac{k_1+1}{N}\right) \times \left(\frac{k_2}{N}, \frac{k_2+1}{N}\right) \times \cdots \times \left(\frac{k_d}{N}, \frac{k_d+1}{N}\right),
$$

$$
\Omega_k^{in} = \left[\frac{k_1}{N} + \delta, \frac{k_1+1}{N} - \delta\right] \times \left[\frac{k_2}{N} + \delta, \frac{k_2+1}{N} - \delta\right] \times \cdots \times \left[\frac{k_d}{N} + \delta, \frac{k_d+1}{N} - \delta\right].
$$

For $x \in \Omega_k^{in}$, we immediately obtain a lower bound of $\sum_{i=1}^d g_{k_i}^{(1)}(x_i)$ from Lemma C.1.

**Corollary C.1.** *When $x = (x_1, \cdots, x_d) \in \Omega_k^{in}$,*

$$
\sum_{i=1}^d g_{k_i}^{(1)}(x_i) \geq d - \frac{2d}{1+\ln\alpha}.
$$

For $x \in \Omega \setminus \Omega_k^{out}$, we can derive an upper bound of $\sum_{i=1}^{d} g_{k_i}^{(1)}(x_i)$ from Lemma C.1.

**Lemma C.2.** *Assume* $\alpha > e^{2d+1}$. *When* $x = (x_1, \cdots, x_d) \in \Omega \setminus \Omega_k^{out}$,

$$\sum_{i=1}^{d} g_{k_i}^{(1)}(x_i) \le d - \frac{2d+1}{1 + \ln \alpha}.$$

*Proof.* We prove the lemma by contradiction. Assume that there exists $x \in \Omega \setminus \Omega_k^{out}$ such that $\sum_{i=1}^{d} g_{k_i}^{(1)}(x_i) > d - \frac{2d+1}{1+\ln \alpha}$. Since $x \in \Omega \setminus \Omega_k^{out}$, by Lemma C.1 there exists $1 \le j \le d$ such that $g_{k_j}^{(1)}(x_i) < \frac{1}{1+\ln \alpha}$, which implies

$$\sum_{i=1, i \ne j}^{d} g_{k_i}^{(1)}(x_i) > d - \frac{2d+1}{1 + \ln \alpha} - \frac{1}{1 + \ln n} = d - \frac{2d+2}{1 + \ln \alpha}.$$

But when $\alpha > e^{2d+1}$, we have $d - \frac{2d+2}{1+\ln \alpha} > d - 1$, which is contradict with the fact from the fact that $g_k^{(1}(x_i) < 1$ for any $1 \le i \le d$.

$\square$

Define

$$g^{(2)}(x) = \sigma \left[ (2 \ln^2 \alpha + 2 \ln \alpha) x - 2d \ln^2 \alpha + (2d+1) \ln \alpha \right]. \tag{C.2}$$

Define grid points:

$$x^{(k)} = \left( \frac{k_1 + \frac{1}{2}}{N}, \frac{k_2 + \frac{1}{2}}{N}, \cdots, \frac{k_d + \frac{1}{2}}{N} \right), \quad k \in [0, 1, \cdots, N-1]^d.$$

The following lemma shows that if we set the value of $N, \delta, \alpha$ properly, we can construct an approximator of $f$ based on $\left\{ g_{k_i}^{(1)} \right\}_{i=1}^{d}$ and $g^{(2)}$. Here we establish the approximation result in $L^4$ space.

**Lemma C.3.** *For any* $f : [0,1]^d \to \mathbb{R}$ *and any* $\epsilon > 0$, *set*

$$N = \left\lceil \frac{2\sqrt{d} L_f}{\epsilon} \right\rceil, \quad \alpha = \max \left\{ \frac{4 B_f}{\epsilon} \left( \left\lceil \frac{2\sqrt{d} L_f}{\epsilon} \right\rceil \right)^d - 1, e^{2d+1} \right\}, \quad \delta = \frac{15 \epsilon^4}{2d(\epsilon + 4)^4} \frac{1}{\left\lceil \frac{4\sqrt{d} L_f}{\epsilon} \right\rceil}$$

*in (C.1) and (C.2). Then the function*

$$g(x) := \sum_{k \in \{0, 1, \cdots, N-1\}^d} f\left( x^{(k)} \right) g^{(2)} \left( \sum_{i=1}^{d} g_{k_i}^{(1)}(x_i) \right)$$

*satisfies*

$$\|f - g\|_4 \le \epsilon,$$
$$\|f - g\|_\infty \le 2 + \frac{\epsilon}{2}.$$

*Proof.* Rewrite $f$ as(Except for the boundary point)

$$f(x) = \sum_{k \in \{0, 1, \cdots, N-1\}^d} f(x) \mathbb{1}_k(x),$$

where

$$\mathbb{1}_k(x) = \begin{cases} 1, & x \in \Omega_k^{out} \\ 0, & x \notin \Omega_k^{out} \end{cases}.$$

We estimate the difference of $f$ and $g$:

$$
\begin{aligned}
|g(x) - f(x)| &= \left| \sum_{k \in \{0,1,\cdots,N-1\}^d} \left[ f\left(x^{(k)}\right) g^{(2)}\left(\sum_{i=1}^d g_{k_i}^{(1)}(x_i)\right) - f(x) \mathbb{1}_k(x) \right] \right| \\
&\leq \sum_{k \in \{0,1,\cdots,N-1\}^d} \left| f\left(x^{(k)}\right) \right| \left| g^{(2)}\left(\sum_{i=1}^d g_{k_i}^{(1)}(x_i)\right) - \mathbb{1}_k(x) \right| + \sum_{k \in \{0,1,\cdots,N-1\}^d} \left| f\left(x^{(k)}\right) - f(x) \right| |\mathbb{1}_k(x)| \\
&\leq B_f \sum_{k \in \{0,1,\cdots,N-1\}^d} \left| g^{(2)}\left(\sum_{i=1}^d g_{k_i}^{(1)}(x_i)\right) - \mathbb{1}_k(x) \right| + L_f \sum_{k \in \{0,1,\cdots,N-1\}^d} \|x^{(k)} - x\|_2 \cdot \mathbb{1}_k(x) \\
&\leq B_f \sum_{k \in \{0,1,\cdots,N-1\}^d} \left| g^{(2)}\left(\sum_{i=1}^d g_{k_i}^{(1)}(x_i)\right) - \mathbb{1}_k(x) \right| + \frac{\sqrt{d} L_f}{2N}.
\end{aligned}
$$

Now we study $\left| g^{(2)}\left(\sum_{i=1}^d g_{k_i}^{(1)}(x_i)\right) - \mathbb{1}_k(x) \right|$, that is, the difference of $g^{(2)}$ and the indicator function. When $x \in \Omega_k^{in}$, by Corollary C.1 we have

$$
1 - \frac{1}{1+\alpha} = \sigma(\ln \alpha) = g^{(2)}\left(d - \frac{2d}{1 + \ln \alpha}\right) \leq g^{(2)}\left(\sum_{i=1}^d g_{k_i}^{(1)}(x_i)\right) < 1.
$$

Hence

$$
\left| g^{(2)}\left(\sum_{i=1}^d g_{k_i}^{(1)}(x_i)\right) - \mathbb{1}_k(x) \right| = \left| g^{(2)}\left(\sum_{i=1}^d g_{k_i}^{(1)}(x_i)\right) - 1 \right| \leq \frac{1}{1+\alpha}.
$$

When $x \in \Omega \setminus \Omega_k^{out}$, by Lemma C.2 we have

$$
0 < g^{(2)}\left(\sum_{i=1}^d g_{k_i}^{(1)}(x_i)\right) \leq g^{(2)}\left(d - \frac{2d+1}{1 + \ln \alpha}\right) = \sigma(-\ln \alpha) = \frac{1}{1+\alpha}.
$$

Hence

$$
\left| g^{(2)}\left(\sum_{i=1}^d g_{k_i}^{(1)}(x_i)\right) - \mathbb{1}_k(x) \right| = \left| g^{(2)}\left(\sum_{i=1}^d g_{k_i}^{(1)}(x_i)\right) \right| \leq \frac{1}{1+\alpha}.
$$

When $x \in \Omega_k^{out} \setminus \Omega_k^{in}$, we just need a finite upper bound.

$$
\left| g^{(2)}\left(\sum_{i=1}^d g_{k_i}^{(1)}(x_i)\right) - \mathbb{1}_k(x) \right| \leq \left| g^{(2)}\left(\sum_{i=1}^d g_{k_i}^{(1)}(x_i)\right) \right| + |\mathbb{1}_k(x)| < 2.
$$

Therefore, for $x \in \bigcup_{k \in \{0,1,\cdots,N\}^d} \Omega_k^{in}$,

$$
|g(x) - f(x)| \leq \frac{B_f N^d}{1+\alpha} + \frac{\sqrt{d} L_f}{2N};
$$

for $x \in \Omega \setminus \left( \bigcup_{k \in \{0,1,\cdots,N\}^d} \Omega_k^{in} \right)$,

$$
|g(x) - f(x)| \leq \frac{B_f(N^d - 1)}{1+\alpha} + 2 + \frac{\sqrt{d} L_f}{2N}.
$$

Hence

$$
\begin{aligned}
\int_\Omega |g(x) - f(x)|^4 dx &= \int_{\bigcup_{k \in \{0,1,\cdots,N\}^d} \Omega_k^{in}} |g(x) - f(x)|^4 dx + \int_{\Omega \setminus \bigcup_{k \in \{0,1,\cdots,N\}^d} \Omega_k^{in}} |g(x) - f(x)|^4 dx \\
&\leq (1 - 2N\delta)^d \left( \frac{B_f N^d}{1+\alpha} + \frac{\sqrt{d} L_f}{2N} \right)^4 + [1 - (1 - 2N\delta)^d] \left( \frac{B_f(N^d - 1)}{1+\alpha} + 2 + \frac{\sqrt{d} L_f}{2N} \right)^4
\end{aligned}
$$

$$\leq \left(\frac{B_f N^d}{1+\alpha} + \frac{\sqrt{d}L_f}{2N}\right)^4 + 2dN\delta \left(\frac{B_f(N-1)^d}{1+\alpha} + 2 + \frac{\sqrt{d}L_f}{2N}\right)^4,$$

where the third step is due to Bernoulli's inequality(provided $N\delta < \frac{1}{2}$). By our choice of $N, \delta, \alpha$, we have

$$\frac{B_f N^d}{1+\alpha} \leq \frac{\epsilon}{4}, \quad \frac{\sqrt{d}L_f}{2N} \leq \frac{\epsilon}{4}, \quad 2dN\delta \left(\frac{B_f(N^d-1)}{1+\alpha} + 2 + \frac{\sqrt{d}L_f}{2N}\right)^4 \leq 2dN\delta \left(\frac{1}{2}\epsilon + 2\right)^4 \leq \frac{15\epsilon^4}{16},$$

which completing the proof. $\qquad\square$

Let $v = (1, \cdots, 1) \in \mathbb{R}^d$.

**Lemma C.4.** *For any $f : [-B_x, B_x]^d \to \mathbb{R}$ and any $\epsilon > 0$, set*

$$N = \left\lceil \frac{2^{d/4+1}\sqrt{d}B_x^{d/4}L_f}{\epsilon} \right\rceil, \quad \delta = \frac{15\epsilon^4}{2d\left(\epsilon + 2^{d/4+2}B_x^{d/4}\right)^4} \frac{1}{\left\lceil \frac{2^{d/4+1}\sqrt{d}B_x^{d/4}L_f}{\epsilon} \right\rceil},$$

$$\alpha = \max\left\{ \frac{2^{d/4+2}B_x^{d/4}B_f}{\epsilon} \left(\left\lceil \frac{2^{d/4+1}\sqrt{d}B_x^{d/4}L_f}{\epsilon} \right\rceil\right)^d - 1, e^{2d+1} \right\}$$

*in (C.1) and (C.2). Then the function*

$$g(x) := \sum_{k \in \{0,1,\cdots,N-1\}^d} f\left(2B_x x^{(k)} - B_x v\right) g^{(2)}\left(\sum_{i=1}^d g_{k_i}^{(1)}\left(\frac{1}{2B_x}x_i + \frac{1}{2}\right)\right)$$

*satisfies*

$$\|f - g\|_4 \leq \epsilon,$$
$$\|f - g\|_\infty \leq 2 + \frac{\epsilon}{2^{d/4+1}B_x^{d/4}}$$

*Proof.* Define $\widetilde{f}(x) := f(2B_x x - B_x v)$ and

$$\widetilde{g}(x) := g(2B_x x - B_x v) = \sum_{k \in \{0,1,\cdots,N-1\}^d} f\left(2B_x x^{(k)} - B_x v\right) g^{(2)}\left(\sum_{i=1}^d g_{k_i}^{(1)}(x_i)\right)$$

$$= \sum_{k \in \{0,1,\cdots,N-1\}^d} \widetilde{f}\left(x^{(k)}\right) g^{(2)}\left(\sum_{i=1}^d g_{k_i}^{(1)}(x_i)\right)$$

then the domains of $\widetilde{f}$ and $\widetilde{g}$ are both $[0,1]^d$. From Lemma C.3 we have $\left\|\widetilde{f} - \widetilde{g}\right\|_4 \leq \frac{\epsilon}{2^{d/4}B_x^{d/4}}$ and $\left\|\widetilde{f} - \widetilde{g}\right\|_\infty \leq 2 + \frac{\epsilon}{2^{d/4+1}B_x^{d/4}}$. Using the relations $f(x) = \widetilde{f}\left(\frac{1}{2B_x}x + \frac{1}{2}v\right), g(x) = \widetilde{g}\left(\frac{1}{2B_x}x + \frac{1}{2}v\right)$, we have $|f(x) - g(x)| = \left|\widetilde{f}\left(\frac{1}{2B_x}x + \frac{1}{2}v\right) - \widetilde{g}\left(\frac{1}{2B_x}x + \frac{1}{2}v\right)\right| \leq 2 + \frac{\epsilon}{2^{d/4+1}B_x^{d/4}}$ and

$$\int_{x \in [-B_x, B_x]^d} |f(x) - g(x)|^4 dx = \int_{x \in [-B_x, B_x]^d} \left|\widetilde{f}\left(\frac{1}{2B_x}x + \frac{1}{2}v\right) - \widetilde{g}\left(\frac{1}{2B_x}x + \frac{1}{2}v\right)\right|^4 dx$$

$$= 2^d B_x^d \int_{t \in [0,1]^d} |\widetilde{f}(t) - \widetilde{g}(t)|^4 dt \leq 2^d B_x^d \cdot \frac{\epsilon^4}{2^d B_x^d} = \epsilon^4.$$

$\qquad\square$

The width of the approximator $g$ in Lemma C.4 is $dN^d$. However, when we perform an optimization algorithm(e.g., GD), the width of the neural network in the function class we choose may exceed $dN^d$. Recall that our $\mathcal{F}_{NN}$ is defined by

$$\mathcal{F}_{NN}(\{B_3, B_2, B_1\}, m_2) := \left\{ \frac{1}{\sqrt{m_2}} \sum_{k_2=1}^{m_2} w_{k_2}^{(3)} \sigma \left( \sum_{k_1 \in I_{k_2}^{(2)}} w_{k_2 k_1}^{(2)} \sigma \left( \sum_{k_0=1}^{d} w_{k_1 k_0}^{(1)} x_{k_0} + b_{k_1}^{(1)} \right) + b_{k_2}^{(2)} \right) : \right.$$
$$\left. \left| w_{k_2}^{(3)} \right| \le B_3, \left| w_{k_2 k_1}^{(2)} \right|, \left| b_{k_2}^{(2)} \right| \le B_2, \left| w_{k_1 k_0}^{(1)} \right|, \left| b_{k_1}^{(1)} \right| \le B_1 \right\}.$$

The width of neural network functions in this class is $m_2 d$ with $m_2$ being an extremely large number(hence $m_2 > N^d$). An important issue is to make sure that there is still an approximator lying in $\mathcal{F}_{NN}$. Fortunately, $g$ still belongs to $\mathcal{F}_{NN}$. To see this, we only need to widen $g$ by just setting 0-value weights in the added neuron units. Specifically, we can express $g$ in the form of functions in $\mathcal{F}_{NN}$ by letting $w_{k_2}^{(3)} = 0, w_{k_2 k_1}^{(2)} = 0$ for $k_2 > N^d$. Hence based on Lemma C.4, we obtain Theorem 2.

# D    PROOF OF THEOREM 3: GENERALIZATION ERROR AND RADEMACHER COMPLEXITY

In this section, we study the generalization error by the covering number. To derive an upper bound for the covering number of the neural network function class, we follow the path in Li & Ding (2021)Drews & Kohler (2022). But compared with Drews & Kohler (2022), we give an induction proof of the formula of the higher-order derivative(with respect to the weights) of the neural network function and derive an upper bound with explicit independence on all the parameters (Lemma D.4).

Define $\mathcal{F}_1 := \left\{ [f(X) - Y]^2 : f \in \mathcal{F}_{NN}^{(bounded)} \right\}$.

**Lemma D.1.** *We have*

$$\mathbb{E}_{\{Z_i\}} \sup_{f \in \mathcal{F}_{NN}^{(bounded)}} \left[ \frac{1}{n} \sum_{i=1}^{n} [f(X_i) - Y_i]^2 - \mathbb{E}[f(X) - Y]^2 \right] \le 2\mathfrak{R}_n(\mathcal{F}_1),$$

$$\mathbb{E}_{\{Z_i\}} \sup_{f \in \mathcal{F}_{NN}^{(bounded)}} \left[ \mathbb{E}[f(X) - Y]^2 - \frac{1}{n} \sum_{i=1}^{n} [f(X_i) - Y_i]^2 \right] \le 2\mathfrak{R}_n(\mathcal{F}_1).$$

*Proof.* We only prove the first inequality and the second one can be proved similarly. Let $\{Z_i\}_{i=1}^{n}$ be an i.i.d. copy of $\{Z_i\}_{i=1}^{n}$. We have

$$\frac{1}{n} \sum_{i=1}^{n} [f(X_i) - Y_i]^2 - \mathbb{E}[f(X) - Y]^2 = \frac{1}{n} \mathbb{E}_{\{\widetilde{Z}_i\}} \sum_{i=1}^{n} \left[ [f(X_i) - Y_i]^2 - [f(\widetilde{X}_i) - \widetilde{Y}_i]^2 \right].$$

then

$$\mathbb{E}_{\{Z_i\}} \sup_{f \in \mathcal{F}_{NN}^{(bounded)}} \left[ \frac{1}{n} \sum_{i=1}^{n} [f(X_i) - Y_i]^2 - \mathbb{E}[f(X) - Y]^2 \right]$$

$$= \frac{1}{n} \mathbb{E}_{\{Z_i\}} \sup_{f \in \mathcal{F}_{NN}^{(bounded)}} \mathbb{E}_{\{\widetilde{Z}_i\}} \sum_{i=1}^{n} \left[ [f(X_i) - Y_i]^2 - [f(\widetilde{X}_i) - \widetilde{Y}_i]^2 \right]$$

$$\le \frac{1}{n} \mathbb{E}_{\{\widetilde{Z}_i, Z_i\}} \sup_{f \in \mathcal{F}_{NN}^{(bounded)}} \sum_{i=1}^{n} \left[ [f(X_i) - Y_i]^2 - [f(\widetilde{X}_i) - \widetilde{Y}_i]^2 \right]$$

$$= \frac{1}{n} \mathbb{E}_{\{\widetilde{Z}_i, Z_i, \varsigma_i\}} \sup_{f \in \mathcal{F}_{NN}^{(bounded)}} \sum_{i=1}^{n} \varsigma_i \left[ [f(X_i) - Y_i]^2 - [f(\widetilde{X}_i) - \widetilde{Y}_i]^2 \right]$$

$$\le \frac{1}{n} \mathbb{E}_{\{Z_i, \varsigma_i\}} \sup_{f \in \mathcal{F}_{NN}^{(bounded)}} \sum_{i=1}^{n} \varsigma_i [f(X_i) - Y_i]^2 + \frac{1}{n} \mathbb{E}_{\{\widetilde{Z}_i, \varsigma_i\}} \sup_{f \in \mathcal{F}} \sum_{i=1}^{n} (-\varsigma_i) [f(\widetilde{X}_i) - \widetilde{Y}_i]^2$$

$$= \frac{2}{n}\mathbb{E}_{\{Z_i,\varsigma_i\}}\sup_{f\in\mathcal{F}_{NN}^{(bounded)}}\sum_{i=1}^{n}\varsigma_i[f(X_i)-Y_i]^2 = 2\mathfrak{R}_n(\mathcal{F}_1),$$

where the third step is due to the fact that the insertion of Rademacher variables $\{\varsigma_i\}$ doesn't change the distribution. $\qquad\square$

We can bound the Rademacher complexity by the covering number. The following lemma appears in Jiao et al. (2021).

**Lemma D.2.** *For any function class $\mathcal{G}$ with $|g| \le B_{\mathcal{G}}$ for all $g \in \mathcal{G}$,*

$$\mathfrak{R}_n(\mathcal{G}) \le \inf_{0<\delta<B_{\mathcal{G}}/2}\left(4\delta + \frac{12}{\sqrt{n}}\int_{\delta}^{B_{\mathcal{G}}/2}\sqrt{\ln\mathcal{N}(\epsilon,\mathcal{G},\|\cdot\|_\infty,n)}d\epsilon\right).$$

Since $\mathcal{F}_1$ generates from $\mathcal{F}_{NN}^{(bounded)}$, the following lemma shows we only need to study the coveirng number of $\mathcal{F}_{NN}^{(bounded)}$.

**Lemma D.3.** *Assume $\epsilon < B_{tr}$. We have*

$$\mathcal{N}((3B_{tr}+2B_Y)\epsilon,\mathcal{F}_1,\|\cdot\|_\infty,n) \le \mathcal{N}\left(\epsilon,\mathcal{F}_{NN}^{(bounded)},\|\cdot\|_\infty,n\right).$$

*Proof.* Assume $\mathcal{C}\left(\epsilon,\mathcal{F}_{NN}^{(bounded)}|_{\{X_i\}_{i=1}^n},\|\cdot\|_\infty\right)$ is an $\epsilon$-covering of $\mathcal{F}_{NN}^{(bounded)}|_{\{X_i\}_{i=1}^n}$ with the least caradinality. For any $f \in \mathcal{F}_{NN}^{(bounded)}$, choose $\bar{f} \in \mathcal{C}(\epsilon,\mathcal{F}_{NN}^{(bounded)}|_{\{X_i\}_{i=1}^n},\|\cdot\|_\infty)$(and hence $\bar{f}$ is a $n$-dimensional vector and we denote its $i$th component as $\bar{f}_i$) such that $\|f|_{\{X_i\}_{i=1}^n} - \bar{f}\|_\infty \le \epsilon$. Then for $i \in [n]$,

$$\left|[f(X_i)-Y_i]^2-[\bar{f}_i-Y_i]^2\right| = |\bar{f}_i+f(X_i)-2Y_i||\bar{f}_i-f(X_i)| \le (3B_{tr}+2B_Y)\epsilon.$$

Hence $\left\{(\bar{f}-Y)^2 : \bar{f} \in \mathcal{C}\left(\epsilon,\mathcal{F}_{NN}^{(bounded)}|_{\{X_i\}_{i=1}^n},\|\cdot\|_\infty\right)\right\}$ is an $(3B_{tr}+2B_Y)\epsilon$-covering of $\mathcal{F}_1|_{\{Z_i\}_{i=1}^n}$(where $(\bar{f}-Y)^2$ refers to the vector $((\bar{f}_1-Y_1)^2,\cdots,(\bar{f}_n-Y_n)^2)$ and we denote it as $\mathcal{C}((3B_{tr}+2B_Y)\epsilon,\mathcal{F}_1|_{\{Z_i\}_{i=1}^n},\|\cdot\|_\infty)$. Therefore

$$\mathcal{N}((3B_{tr}+2B_Y)\epsilon,\mathcal{F}_1|_{\{Z_i\}_{i=1}^n},\|\cdot\|_\infty) \le \left|\mathcal{C}((3B_{tr}+2B_Y)\epsilon,\mathcal{F}_1|_{\{Z_i\}_{i=1}^n},\|\cdot\|_\infty)\right|$$
$$= \left|\mathcal{C}\left(\epsilon,\mathcal{F}_{NN}^{(bounded)}|_{\{X_i\}_{i=1}^n},\|\cdot\|_\infty\right)\right| = \mathcal{N}\left(\epsilon,\mathcal{F}_{NN}^{(bounded)}|_{\{X_i\}_{i=1}^n},\|\cdot\|_\infty\right)$$

and hence

$$\mathcal{N}((3B_{tr}+2B_Y)\epsilon,\mathcal{F}_1,\|\cdot\|_\infty,n) = \max_{\{Z_i\}_{i=1}^n}\mathcal{N}((3B_{tr}+2B_Y)\epsilon,\mathcal{F}_1|_{\{Z_i\}_{i=1}^n},\|\cdot\|_\infty)$$
$$\le \max_{\{X_i\}_{i=1}^n}\mathcal{N}\left(\epsilon,\mathcal{F}_{NN}^{(bounded)}|_{\{X_i\}_{i=1}^n},\|\cdot\|_\infty\right) = \mathcal{N}\left(\epsilon,\mathcal{F}_{NN}^{(bounded)},\|\cdot\|_\infty,n\right).$$

$\qquad\square$

To calculate the covering number of $\mathcal{F}_{NN}^{(bounded)}$, we need an estimate of the upper bound of high-order derivatives of functions in $\mathcal{F}_{NN}$.

**Lemma D.4.** *Let $f_W \in \mathcal{F}_{NN}(\{B_3,B_2,B_1\},m_2)$. For any $j_1,\cdots,j_d \in \mathbb{N}_{\ge 0}$,*

$$\left|\frac{\partial^j f_W}{\partial x_1^{j_1}\partial x_2^{j_2}\cdots\partial x_d^{j_d}}\right| \le \sqrt{m_2}\widetilde{m}_2^j B_3 B_2^j B_1^j \max_{j_1\in\{1,2,\cdots,j\}}B_{\sigma^{(j_1)}}^{j+1}\mathcal{B}_j,$$

*where $j = j_1 + \cdots + j_d$.*

*Proof.* By definition,

$$\frac{\partial^j f_W}{\partial x_1^{j_1}\partial x_2^{j_2}\cdots\partial x_d^{j_d}} = \frac{1}{\sqrt{m_2}}\sum_{k_2=1}^{m_2}w_{k_2}^{(3)}\frac{\partial^j f_{k_2}^{(2)}}{\partial x_1^{j_1}\partial x_2^{j_2}\cdots\partial x_d^{j_d}}. \qquad\text{(D.1)}$$

We rewrite $\frac{\partial^j f_W}{\partial x_1^{j_1} \partial x_2^{j_2} \cdots \partial x_d^{j_d}}$ as $\frac{\partial^j f_{k_2}^{(2)}}{\partial x_{p_1} \cdots \partial x_{p_j}} (p_1, \cdots, p_j \in [d])$ and prove the following formula of the derivative of $f_{k_2}^{(2)}$ by induction:

$$\frac{\partial^j f_{k_2}^{(2)}}{\partial x_{p_1} \cdots \partial x_{p_j}} = \sum_{q=1}^{j} \sigma^{(q)}(f_{2,k_2}^{org}) \sum_{\substack{\alpha^{(1)}+\cdots+\alpha^{(q)}=(1,\cdots,1) \\ \alpha^{(i_1)}>\alpha^{(i_2)}>0,\ 1\le i_1<i_2\le q}} \prod_{i=1}^{q} \left( \sum_{k_1\in I_{m_2}^{(2)}} w_{k_2 k_1}^{(2)} \frac{\partial^{|\alpha^{(i)}|} f_{k_1}^{(1)}}{\partial x_{p_1}^{\alpha_1^{(i)}} \cdots \partial x_{p_j}^{\alpha_j^{(i)}}} \right),$$

(D.2)

where $\alpha^{(i_2)} > 0$ means that $|\alpha^{(i_2)}| > 0$ and $\alpha^{(i_1)} > \alpha^{(i_2)}$ means there exists $1 \le i' \le q$ such that for $1 \le i'' < i'$, $\alpha_{i''}^{(i_1)} = \alpha_{i''}^{(i_2)}$ and $\alpha_{i'}^{(i_1)} = 1, \alpha_{i'}^{(i_2)} = 0$. Assume the formula holds for $j$ and we prove the case for $j + 1$:

$$\frac{\partial^{j+1} f_{k_2}^{(2)}}{\partial x_{p_1} \cdots \partial x_{p_j} \partial x_{p_{j+1}}} = \frac{\partial}{\partial x_{p_{j+1}}} \frac{\partial^j f_{k_2}^{(2)}}{\partial x_{p_1} \cdots \partial x_{p_j}}$$

$$= \sum_{q=1}^{j} \frac{\partial \sigma^{(q)}(f_{2,k_2}^{org})}{\partial x_{p_{j+1}}} \sum_{\substack{\alpha^{(1)}+\cdots+\alpha^{(q)}=(1,\cdots,1) \\ \alpha^{(i_1)}>\alpha^{(i_2)}>0,\ 1\le i_1<i_2\le q}} \prod_{i=1}^{q} \left( \sum_{k_1\in I_{m_2}^{(2)}} w_{k_2 k_1}^{(2)} \frac{\partial^{|\alpha^{(i)}|} f_{k_1}^{(1)}}{\partial x_{p_1}^{\alpha_1^{(i)}} \cdots \partial x_{p_j}^{\alpha_j^{(i)}}} \right)$$

$$+ \sum_{q=1}^{j} \sigma^{(q)}(f_{2,k_2}^{org}) \sum_{\substack{\alpha^{(1)}+\cdots+\alpha^{(q)}=(1,\cdots,1) \\ \alpha^{(i_1)}>\alpha^{(i_2)}>0,\ 1\le i_1<i_2\le q}} \frac{\partial}{\partial x_{p_{j+1}}} \prod_{i=1}^{q} \left( \sum_{k_1\in I_{m_2}^{(2)}} w_{k_2 k_1}^{(2)} \frac{\partial^{|\alpha^{(i)}|} f_{k_1}^{(1)}}{\partial x_{p_1}^{\alpha_1^{(i)}} \cdots \partial x_{p_j}^{\alpha_j^{(i)}}} \right)$$

$$= \sum_{q=1}^{j} \sigma^{(q+1)}(f_{2,k_2}^{org}) \sum_{k_1\in I_{m_2}^{(2)}} w_{k_2 k_1}^{(2)} \frac{\partial f_{k_1}^{(1)}}{\partial x_{p_{j+1}}} \sum_{\substack{\alpha^{(1)}+\cdots+\alpha^{(q)}=(1,\cdots,1) \\ \alpha^{(i_1)}>\alpha^{(i_2)}>0,\ 1\le i_1<i_2\le q}} \prod_{i=1}^{q} \left( \sum_{k_1\in I_{m_2}^{(2)}} w_{k_2 k_1}^{(2)} \frac{\partial^{|\alpha^{(i)}|} f_{k_1}^{(1)}}{\partial x_{p_1}^{\alpha_1^{(i)}} \cdots \partial x_{p_j}^{\alpha_j^{(i)}}} \right)$$

$$+ \sum_{q=1}^{j} \sigma^{(q)}(f_{2,k_2}^{org}) \sum_{\substack{\alpha^{(1)}+\cdots+\alpha^{(q)}=(1,\cdots,1) \\ \alpha^{(i_1)}>\alpha^{(i_2)}>0,\ 1\le i_1<i_2\le q}} \sum_{\substack{\alpha_{j+1}^{(1)}+\cdots+\alpha_{j+1}^{(q)}=1 \\ \alpha_{j+1}^{(i)}\ge 0,\ 1\le i\le q}} \prod_{i=1}^{q} \left( \sum_{k_1\in I_{m_2}^{(2)}} w_{k_2 k_1}^{(2)} \frac{\partial^{|\alpha^{(i)}|} f_{k_1}^{(1)}}{\partial x_{p_1}^{\alpha_1^{(i)}} \cdots \partial x_{p_j}^{\alpha_j^{(i)}} \partial x_{p_{j+1}}^{\alpha_{j+1}^{(i)}}} \right)$$

$$= \sum_{q=1}^{j} \sigma^{(q+1)}(f_{2,k_2}^{org}) \sum_{\substack{\alpha^{(1)}+\cdots+\alpha^{(q)}=(1,\cdots,1) \\ \alpha^{(i_1)}>\alpha^{(i_2)}>0,\ 1\le i_1<i_2\le q}} \prod_{i=1}^{q} \left( \sum_{k_1\in I_{m_2}^{(2)}} w_{k_2 k_1}^{(2)} \frac{\partial^{|\alpha^{(i)}|} f_{k_1}^{(1)}}{\partial x_{p_1}^{\alpha_1^{(i)}} \cdots \partial x_{p_j}^{\alpha_j^{(i)}} \partial x_{p_{j+1}}^{0}} \right)$$

$$\left( \sum_{k_1\in I_{m_2}^{(2)}} w_{k_2 k_1}^{(2)} \frac{\partial f_{k_1}^{(1)}}{\partial x_{p_1}^{0} \cdots \partial x_{p_j}^{0} \partial x_{p_{j+1}}^{1}} \right)$$

$$+ \sum_{q=1}^{j} \sigma^{(q)}(f_{2,k_2}^{org}) \sum_{\substack{\bar{\alpha}^{(1)}+\cdots+\bar{\alpha}^{(q)}=(1,\cdots,1,1) \\ \bar{\alpha}^{(q)}\ne(0,\cdots,0,1) \\ \alpha^{(i_1)}>\alpha^{(i_2)}>0,\ 1\le i_1<i_2\le q}} \prod_{i=1}^{q} \left( \sum_{k_1\in I_{m_2}^{(2)}} w_{k_2 k_1}^{(2)} \frac{\partial^{|\alpha^{(i)}|} f_{k_1}^{(1)}}{\partial x_{p_1}^{\alpha_1^{(i)}} \cdots \partial x_{p_j}^{\alpha_j^{(i)}} \partial x_{p_{j+1}}^{\alpha_{j+1}^{(i)}}} \right)$$

$$= \sum_{q=1}^{j} \sigma^{(q+1)}(f_{2,k_2}^{org}) \sum_{\substack{\bar{\alpha}^{(1)}+\cdots+\bar{\alpha}^{(q)}=(1,\cdots,1,0) \\ \bar{\alpha}^{(q+1)}=(0,\cdots,0,1) \\ \bar{\alpha}^{(i_1)}>\bar{\alpha}^{(i_2)}>0,\ 1\le i_1<i_2\le q+1}} \prod_{i=1}^{q+1} \left( \sum_{k_1\in I_{m_2}^{(2)}} w_{k_2 k_1}^{(2)} \frac{\partial^{|\bar{\alpha}^{(i)}|} f_{k_1}^{(1)}}{\partial x_{p_1}^{\alpha_1^{(i)}} \cdots \partial x_{p_j}^{\alpha_j^{(i)}} \partial x_{p_{j+1}}^{\alpha_{j+1}^{(i)}}} \right)$$

$$+ \sum_{q=1}^{j} \sigma^{(q)}(f_{2,k_2}^{org}) \sum_{\substack{\bar{\alpha}^{(1)}+\cdots+\bar{\alpha}^{(q)}=(1,\cdots,1,1) \\ \bar{\alpha}^{(q)} \neq (0,\cdots,0,1) \\ \alpha^{(i_1)} > \alpha^{(i_2)} > 0, \, 1 \leq i_1 < i_2 \leq q}} \prod_{i=1}^{q} \left( \sum_{k_1 \in I_{m_2}^{(2)}} w_{k_2 k_1}^{(2)} \frac{\partial^{|\alpha^{(i)}|} f_{k_1}^{(1)}}{\partial x_{p_1}^{\alpha_1^{(i)}} \cdots \partial x_{p_j}^{\alpha_j^{(i)}} \partial x_{p_{j+1}}^{\alpha_{j+1}^{(i)}}} \right)$$

$$= \sum_{q=1}^{j+1} \sigma^{(q)}(f_{2,k_2}^{org}) \sum_{\substack{\bar{\alpha}^{(1)}+\cdots+\bar{\alpha}^{(q)}=(1,\cdots,1,1) \\ \alpha^{(i_1)} > \alpha^{(i_2)} > 0, \, 1 \leq i_1 < i_2 \leq q}} \prod_{i=1}^{q} \left( \sum_{k_1 \in I_{m_2}^{(2)}} w_{k_2 k_1}^{(2)} \frac{\partial^{|\alpha^{(i)}|} f_{k_1}^{(1)}}{\partial x_{p_1}^{\alpha_1^{(i)}} \cdots \partial x_{p_j}^{\alpha_j^{(i)}} \partial x_{p_{j+1}}^{\alpha_{j+1}^{(i)}}} \right) \cdot$$

For the derivative of $f_{k_1}^{(1)}$,

$$\frac{\partial^{|\alpha^{(i)}|} f_{k_1}^{(1)}}{\partial x_{p_1}^{\alpha_1^{(i)}} \cdots \partial x_{p_j}^{\alpha_j^{(i)}}} = \sigma^{|\alpha^{(i)}|}(f_{1,k_1}^{org}) \prod_{j'=1}^{j} \left( w_{k_1 p_{j'}}^{(1)} \right)^{\alpha_{j'}^{(i)}},$$

hence

$$\left| \frac{\partial^{|\alpha^{(i)}|} f_{k_1}^{(1)}}{\partial x_{p_1}^{\alpha_1^{(i)}} \cdots \partial x_{p_j}^{\alpha_j^{(i)}}} \right| \leq B_{\sigma^{(|\alpha^{(i)}|)}} B_1^{|\alpha^{(i)}|}.$$

Plugging the above upper bound into (D.2),

$$\left| \frac{\partial^j f_{k_2}^{(2)}}{\partial x_{p_1} \cdots \partial x_{p_j}} \right| \leq \sum_{q=1}^{j} \left| \sigma^{(q)}(f_{2,k_2}^{org}) \right| \sum_{\substack{\alpha^{(1)}+\cdots+\alpha^{(q)}=(1,\cdots,1) \\ \alpha^{(i_1)} > \alpha^{(i_2)} > 0, \, 1 \leq i_1 < i_2 \leq q}} \prod_{i=1}^{q} \left( \sum_{k_1 \in I_{m_2}^{(2)}} \left| w_{k_2 k_1}^{(2)} \right| \left| \frac{\partial^{|\alpha^{(i)}|} f_{k_1}^{(1)}}{\partial x_{p_1}^{\alpha_1^{(i)}} \cdots \partial x_{p_j}^{\alpha_j^{(i)}}} \right| \right)$$

$$\leq \sum_{q=1}^{j} \widetilde{m}_2^q B_2^q B_1^j B_{\sigma^{(q)}} \sum_{\substack{\alpha^{(1)}+\cdots+\alpha^{(q)}=(1,\cdots,1) \\ \alpha^{(i_1)} > \alpha^{(i_2)} > 0, \, 1 \leq i_1 < i_2 \\ q}} \left( \prod_{i=1}^{q} B_{\sigma^{(|\alpha^{(i)}|)}} \right) \leq \sum_{q=1}^{j} \widetilde{m}_2^q B_2^q B_1^j B_{\sigma^{(q)}} \begin{Bmatrix} j \\ q \end{Bmatrix} \max_{q_1 \in \{1,2,\cdots,q\}} B_{\sigma^{(q_1)}}^q$$

$$\leq \widetilde{m}_2^j B_2^j B_1^j \max_{j_1 \in \{1,2,\cdots,j\}} B_{\sigma^{(j_1)}}^{j+1} \sum_{q=1}^{j} \begin{Bmatrix} j \\ q \end{Bmatrix} \leq \widetilde{m}_2^j B_2^j B_1^j \max_{j_1 \in \{1,2,\cdots,j\}} B_{\sigma^{(j_1)}}^{j+1} \mathcal{B}_j.$$

Here $\begin{Bmatrix} j \\ q \end{Bmatrix}$ is the Stirling number and $\mathcal{B}_j = \sum_{q=0}^{j} \begin{Bmatrix} j \\ q \end{Bmatrix}$ is the Bell number. Plugging the upper bound for derivative of $f_{k_2}^{(2)}$ into (D.1), we obtain

$$\left| \frac{\partial^j f_W}{\partial x_1^{j_1} \partial x_2^{j_2} \cdots \partial x_d^{j_d}} \right| \leq \frac{1}{\sqrt{m_2}} \sum_{k_2=1}^{m_2} \left| w_{k_2}^{(3)} \right| \left| \frac{\partial^j f_{k_2}^{(2)}}{\partial x_1^{j_1} \partial x_2^{j_2} \cdots \partial x_d^{j_d}} \right|$$

$$\leq \sqrt{m_2} \widetilde{m}_2^j B_3 B_2^j B_1^j \max_{j_1 \in \{1,2,\cdots,j\}} B_{\sigma^{(j_1)}}^{j+1} \mathcal{B}_j.$$

$\square$

**Remark 2.** *In Berend & Tassa (2010) an upper bound of Bell number is given:*

$$\mathcal{B}_j < \left( \frac{0.792j}{\ln(j+1)} \right)^j.$$

**Lemma D.5.** *Let $\epsilon > 0$. Let $s \in \mathbb{N}_{>0}$. For*

$$n \geq \binom{s+d-2}{d-1} \left\lceil 2^{\frac{d}{s}} (m_2)^{\frac{d}{2s}} \widetilde{m}_2^d B_x^d B_3^{\frac{d}{s}} B_2^d B_1^d \left( \max_{s_1 \in \{1,2,\cdots,s\}} B_{\sigma^{(s_1)}} \right)^{\frac{(s+1)d}{s}} \mathcal{B}_s^{\frac{d}{s}} \binom{s+d-1}{d-1}^{\frac{d}{s}} \left( \frac{1}{\epsilon} \right)^{\frac{d}{s}} \right\rceil,$$

*we have*

$$\mathcal{N}\left(\epsilon, \mathcal{F}_{NN}^{(bounded)}, \|\cdot\|_\infty, n\right) \leq \left(\frac{2enB_{tr}}{\epsilon \mathrm{Pdim}(f_{tr}\circ\mathcal{T}_\mathcal{F})}\right)^{\mathrm{Pdim}(f_{tr}\circ\mathcal{T}_\mathcal{F})},$$

*where the definition of $\mathcal{T}_\mathcal{F}$ is given in the proof and*

$\mathrm{Pdim}(f_{tr}\circ\mathcal{T}_\mathcal{F}) \leq$

$$\binom{s+d-2}{d-1}\left\lceil 2^{\frac{d}{s}}(m_2)^{\frac{d}{2s}}\widetilde{m}_2^d B_x^d B_3^{\frac{d}{s}} B_2^d B_1^d \left(\max_{s_1\in\{1,2,\cdots,s\}} B_{\sigma^{(s_1)}}\right)^{\frac{(s+1)d}{s}} \mathcal{B}_s^{\frac{d}{s}} \binom{s+d-1}{d-1}^{\frac{d}{s}} \left(\frac{1}{\epsilon}\right)^{\frac{d}{s}} \right\rceil.$$

*Proof.* We estimate the covering number of $\mathcal{F}_{NN}^{(bounded)}$ by the Taylor polynomial. Let $N$ be a parameter defined later. Define grid points $\left\{x^{(k)}\right\}_{k\in\{0,1,\cdots,N-1\}^d}$:

$$x^{(k)} = \left(\frac{2k_1+1-N}{N}B_x, \frac{2k_2+1-N}{N}B_x, \cdots, \frac{2k_d+1-N}{N}B_x\right).$$

Define a series of local Taylor polynomial of $f_W$ with respect to the grid points $x^{(k)}$:

$$T_k(x) = \sum_{|\beta|<s}\frac{D^\beta f_W(x^{(k)})}{\beta!}\left(x-x^{(k)}\right)^\beta, \quad k\in\{0,1,\cdots,N-1\}^d.$$

By multivariate Taylor theorem,

$$f_W(x) - T_k(x) = \sum_{|\beta|=s} R_\beta(x)\left(x-x^{(k)}\right)^\beta,$$

where the remainder

$$|R_\beta(x)| \leq \frac{1}{\beta!}\max_x|D^\beta f_W(x)|.$$

Define the global Taylor polynomial:

$$T_{f_W}(x) = \sum_{k\in\{0,1,\cdots,N-1\}^d} T_k(x)\mathbb{1}_k(x),$$

where

$$\mathbb{1}_k(x) = \begin{cases} 1, & x\in\Omega_k^{out} \\ 0, & x\notin\Omega_k^{out} \end{cases}.$$

Here

$$\Omega_k^{out} := \left(\frac{2k_1-N}{N}B_x, \frac{2k_1+2-N}{N}B_x\right)\times\cdots\times\left(\frac{2k_d-N}{N}B_x, \frac{2k_d+2-N}{N}B_x\right).$$

For any $x\in\Omega$, without loss of generality we assume $x\in\Omega_k^{out}$, then by Lemma D.4,

$$|f_W(x) - T_{f_W}(x)| = |f_W(x) - T_k(x)| \leq \sum_{|\beta|=s}|R_\beta(x)|\left|x-x^{(k)}\right|^\beta$$

$$\leq \sum_{|\beta|=s}\frac{1}{\beta!}\max_x|D^\beta f(x)|\left|x-x^{(k)}\right|^\beta$$

$$\leq \sqrt{m_2}\,\widetilde{m}_2^s B_3 B_2^s B_1^s \max_{s_1\in\{1,2,\cdots,s\}} B_{\sigma^{(s_1)}}^{s+1} \mathcal{B}_s \binom{s+d-1}{d-1}\left(\frac{B_x}{N}\right)^s.$$

Set

$$N = \left\lceil (m_2)^{\frac{1}{2s}}\widetilde{m}_2 B_x B_3^{\frac{1}{s}} B_2 B_1 \left(\max_{s_1\in\{1,2,\cdots,s\}} B_{\sigma^{(s_1)}}\right)^{\frac{s+1}{s}} \mathcal{B}_s^{\frac{1}{s}} \binom{s+d-1}{d-1}^{\frac{1}{s}}\left(\frac{2}{\epsilon}\right)^{\frac{1}{s}} \right\rceil, \quad \text{(D.3)}$$

then

$$|f_{tr} \circ f_W(x) - f_{tr} \circ T_{f_W}(x)| \leq |f_W(x) - T_{f_W}(x)| < \frac{\epsilon}{2}. \tag{D.4}$$

We now compute the covering number of $f_{tr} \circ \mathcal{T}_{\mathcal{F}}$ with $\mathcal{T}_{\mathcal{F}} := \{T_{f_W} : f_W \in \mathcal{F}_{NN}\}$, the set of truncated global Taylor polynomials over $\mathcal{F}_{NN}$. By Theorem 12.2 in Anthony et al. (1999) we have for $n \geq \mathrm{Pdim}(f_{tr} \circ \mathcal{T}_{\mathcal{F}})$,

$$\mathcal{N}\left(\frac{\epsilon}{2}, f_{tr} \circ \mathcal{T}_{\mathcal{F}}, \|\cdot\|_\infty, n\right) \leq \left(\frac{2enB_{tr}}{\epsilon \mathrm{Pdim}(f_{tr} \circ \mathcal{T}_{\mathcal{F}})}\right)^{\mathrm{Pdim}(f_{tr} \circ \mathcal{T}_{\mathcal{F}})}. \tag{D.5}$$

To estimate the pseudo-dimension of $\mathcal{T}_{\mathcal{F}}$, enlarge it to a linear space:

$$\mathcal{T} := \left\{ \sum_{k \in \{0,1,\cdots,N-1\}^d} \sum_{|\beta|<s} a_\beta^{(k)} x^\beta \mathbb{1}_k(x) : a_\beta^{(k)} \in \mathbb{R}, |\beta| < s, k \in \{0,1,\cdots,N-1\}^d \right\}.$$

It is easy to check that $\mathcal{T}_{\mathcal{F}} \subset \mathcal{T}$ and $\mathcal{T}$ is a $\binom{s+d-2}{d-1} N^d$-dimension linear space. By Theorem 11.4 in Anthony et al. (1999) we have

$$\mathrm{Pdim}(\mathcal{T}) = \dim(\mathcal{T}) = \binom{s+d-2}{d-1} N^d.$$

By Theorem 11.3 and Corollary 11.5 in Anthony et al. (1999),

$$\mathrm{Pdim}(f_{tr} \circ \mathcal{T}_{\mathcal{F}}) \leq \mathrm{Pdim}(\mathcal{T}_{\mathcal{F}}) \leq \mathrm{Pdim}(\mathcal{T}) = \binom{s+d-2}{d-1} N^d. \tag{D.6}$$

Combining (D.3)-(D.6) yields the result. $\qquad\square$

Combining Lemma D.1, D.2, D.3 and D.5, we obtain the following upper bound for the generalization error.

**Theorem D.1.** *Let*

$$C_1 = 2^{\frac{1}{4}} (3B_{tr} + 2B_Y)^{\frac{1}{4}} m_2^{\frac{1}{8}} \widetilde{m}_2^d B_x^d B_3^{\frac{1}{4}} B_2^d B_1^d \left(\max_{s_1 \in \{1,2,\cdots,4d\}} B_{\sigma^{(s_1)}}\right)^{\frac{(4d+1)}{4}} \mathcal{B}_{4d}^{\frac{1}{4}} \binom{5d-1}{d-1}^{\frac{1}{4}} \binom{5d-2}{d-1},$$

$$C_2 = \frac{2enB_{tr}(3B_{tr} + 2B_Y)}{C_1}.$$

*Assume* $n \geq \max\left\{n_1^{(low)}, n_2^{(low)}\right\}$ *with*

$$n_1^{(low)} = \frac{C_1}{2B_{tr}^{1/4}(3B_{tr} + 2B_Y)^{1/4}},$$

$$n_2^{(low)} = C_1 \max\left\{ \left(\frac{14}{\ln \frac{2^{3/4}C_2}{(B_{tr}+B_Y)^{3/2}} + \frac{6}{7}}\right)^{2/7}, 1 \right\} \left(\frac{2}{(B_{tr}+B_Y)^2}\right)^{1/4}.$$

*Assume* $5B_{tr}^2 + 2B_{tr}B_Y \geq B_Y^2$. *We have*

$$\left.\begin{array}{l} \mathbb{E}_{\{Z_i\}} \sup_{f \in \mathcal{F}_{NN}^{(bounded)}} \left[\frac{1}{n}\sum_{i=1}^n [f(X_i)-Y_i]^2 - \mathbb{E}[f(X)-Y]^2\right] \\ \mathbb{E}_{\{Z_i\}} \sup_{f \in \mathcal{F}_{NN}^{(bounded)}} \left[\mathbb{E}[f(X)-Y]^2 - \frac{1}{n}\sum_{i=1}^n [f(X_i)-Y_i]^2\right] \end{array}\right\}$$

$$\leq \frac{192}{7} \left(\frac{C_1}{n}\right)^{1/2} \left(\frac{(B_{tr}+B_Y)^2}{2}\right)^{7/8} \left(\ln \frac{2^{3/4}C_2}{(B_{tr}+B_Y)^{3/2}} + \frac{6}{7}\right).$$

*Proof.* Applying Lemma D.3 and Lemma D.5 with $s = 4d$, we have for $\epsilon < B_{tr}(3B_{tr} + 2B_Y)$,

$$\ln \mathcal{N}(\epsilon, \mathcal{F}_1, \|\cdot\|_\infty, n) \leq \ln \mathcal{N}\left(\frac{\epsilon}{3B_{tr} + 2B_Y}, \mathcal{F}_{NN}^{(bounded)}, \|\cdot\|_\infty, n\right)$$

$$\leq \mathrm{Pdim}(\mathcal{T}_{\mathcal{F}}) \ln \left( \frac{(3B_{tr} + 2B_Y)enB_{tr}}{\epsilon \mathrm{Pdim}(\mathcal{T}_{\mathcal{F}})} \right) \leq \frac{C_1}{\epsilon^{1/4}} \ln \frac{C_2}{\epsilon^{3/4}}$$

provided $n \geq C_1 \left( \frac{1}{\epsilon} \right)^{1/4}$. We apply Lemma D.2(our assumption on $B_{\mathcal{F}}$ and $B_Y$ ensures $\frac{B_{\mathcal{F}_1}}{2} \leq \frac{1}{2}(B_{tr} + B_Y)^2 \leq B_{tr}(3B_{tr} + 2B_Y)$, hence the above upper bound of covering number is applicable) to calculate an upper bound of $\mathfrak{R}_n(\mathcal{F}_1)$. By the assumption on $n$, we have $\ln \frac{C_2}{\epsilon^{3/4}} \geq 1$. Therefore provided $n \geq C_1 \left( \frac{1}{\delta} \right)^{1/4}$,

$$\mathfrak{R}_n(\mathcal{F}_1) \leq \inf_{0 < \delta < B_{\mathcal{F}_1}/2} \left( 4\delta + \frac{12}{\sqrt{n}} \int_{\delta}^{B_{\mathcal{F}_1}/2} \sqrt{\ln \mathcal{N}(\epsilon, \mathcal{F}_1, \|\cdot\|_\infty, n)} d\epsilon \right)$$

$$\leq \inf_{0 < \delta < B_{\mathcal{F}_1}/2} \left( 4\delta + \frac{12\sqrt{C_1}}{\sqrt{n}} \int_{\delta}^{B_{\mathcal{F}_1}/2} \frac{1}{\epsilon^{1/8}} \ln^{1/2} \frac{C_2}{\epsilon^{3/4}} d\epsilon \right)$$

$$\leq \inf_{0 < \delta < B_{\mathcal{F}_1}/2} \left( 4\delta + \frac{12\sqrt{C_1}}{\sqrt{n}} \int_{\delta}^{B_{\mathcal{F}_1}/2} \frac{1}{\epsilon^{1/8}} \ln \frac{C_2}{\epsilon^{3/4}} d\epsilon \right)$$

$$= \inf_{0 < \delta < B_{\mathcal{F}_1}/2} \left[ 4\delta + \frac{12\sqrt{C_1}}{\sqrt{n}} \cdot \frac{8}{7} \epsilon^{7/8} \left( \ln \frac{C_2}{\epsilon^{3/4}} + \frac{6}{7} \right) \Bigg|_{\delta}^{B_{\mathcal{F}_1}/2} \right]$$

$$\leq \inf_{0 < \delta < B_{\mathcal{F}_1}/2} \left[ 4\delta + \frac{96}{7} \left( \frac{C_1}{n} \right)^{1/2} \left( \frac{B_{\mathcal{F}_1}}{2} \right)^{7/8} \left( \ln \frac{2^{3/4}C_2}{B_{\mathcal{F}_1}^{3/4}} + \frac{6}{7} \right) \right].$$

Choosing $\delta = \min \left\{ \frac{1}{14} \left( \frac{C_1}{n} \right)^{1/2} \left( \frac{(B_{tr} + B_Y)^2}{2} \right)^{7/8} \left( \ln \frac{2^{3/4}C_2}{(B_{tr}+B_Y)^{3/2}} + \frac{6}{7} \right), \frac{1}{2}(B_{tr} + B_Y)^2 \right\}$, we have

$$\mathfrak{R}_n(\mathcal{F}_1) \leq 14 \left( \frac{C_1}{n} \right)^{1/2} \left( \frac{(B_{tr} + B_Y)^2}{2} \right)^{7/8} \left( \ln \frac{2^{3/4}C_2}{(B_{tr}+B_Y)^{3/2}} + \frac{6}{7} \right),$$

which together with Lemma D.1 implies the result. Note that our choice of $\delta$ and $n$ satisfies the condition $n \geq C_1 \left( \frac{1}{\delta} \right)^{1/4}$. □

Now we can derive Theorem 3 from McDiarmid's inequality.

*Proof of Theorem 3.* Define

$$h(Z_1, \cdots, Z_n) := \sup_{f \in \mathcal{F}_{NN}^{(bounded)}} \left[ \frac{1}{n} \sum_{i=1}^n [f(X_i) - Y_i]^2 - \mathbb{E}[f(X) - Y]^2 \right].$$

We examine the difference of $h$.

$$h(Z_1, \cdots, \widetilde{Z}_i, \cdots, Z_n) - h(Z_1, \cdots, Z_i, \cdots, Z_n)$$
$$\leq \frac{1}{n} \sup_{f \in \mathcal{F}_{NN}^{(bounded)}} \left[ [f(\widetilde{X}_i) - \widetilde{Y}_i]^2 - [f(X_i) - Y_i]^2 \right] \leq \frac{2(B_{tr} + B_Y)^2}{n}.$$

We can bound $h(Z_1, \cdots, Z_i, \cdots, Z_n) - h(Z_1, \cdots, \widetilde{Z}_i, \cdots, Z_n)$ similarly and hence obtain

$$\left| h(Z_1, \cdots, \widetilde{Z}_i, \cdots, Z_n) - h(Z_1, \cdots, Z_i, \cdots, Z_n) \right| \leq \frac{2(B_{tr} + B_Y)^2}{n}.$$

By McDiarmid's inequality we have

$$h - \mathbb{E}h \leq \tau$$

with probability at least $1 - e^{\frac{-n\tau^2}{2(B_{tr}+B_Y)^4}}$, which combining with Theorem D.1 yields the first inequality. The second inequality can be proved in the same way. □

# E   PROOF OF THEOREM 4: CONVERGENCE OF GRADIENT DESCENT UNDER NORM CONSTRAIN

## E.1   SOME EXTRA NOTATIONS

In order to analyse optimization error more convenient, we rewrite functions in $\mathcal{F}_{NN}(\{B_3, B_2, B_1\}, m_2)$ in the following form:

$$f_{k_0}^{(0)} = x_{k_0}, \quad k_0 = 1, \cdots, m_0;$$

$$f_{k_1}^{(1)} = \sigma\left(f_{1,k_1}^{org}\right) = \sigma\left(\sum_{k_0=1}^{m_0} w_{k_1 k_0}^{(1)} f_{k_0}^{(0)} + b_{k_1}^{(1)}\right), \quad k_1 = 1, \cdots, m_1;$$

$$f_{k_2}^{(2)} = \sigma\left(f_{2,k_2}^{org}\right) = \sigma\left(\sum_{k_1 \in I_{k_2}^{(2)}} w_{k_2 k_1}^{(2)} f_{k_1}^{(1)} + b_{k_2}^{(2)}\right), \quad k_2 = 1, \cdots, m_2;$$

$$f_W = f^{(3)} = \frac{1}{\sqrt{m_2}} \sum_{k_2=1}^{m_2} w_{k_2}^{(3)} f_{k_2}^{(2)}.$$

To make derivations and results concise, we introduce some new notations. Let $W^{(1)}, W^{(2)}, W^{(3)}$ be the weight vectors in the first, second and third layer, respectively. Let

$$B_1(W) := \max\left\{\max_{k_0 \in [m_0], k_1 \in [m_1]} \left|w_{k_1 k_0}^{(1)}\right|, \max_{k_1 \in [m_1]} \left|b_{k_1}^{(1)}\right|, 1\right\},$$

$$B_2(W) := \max\left\{\max_{k_1 \in I_{k_2}^{(2)}, k_2 \in [m_2]} \left|w_{k_2 k_1}^{(2)}\right|, \max_{k_2 \in [m_2]} \left|b_{k_2}^{(2)}\right|, 1\right\},$$

$$B_3(W) := \max\left\{\max_{k_2 \in [m_2]} \left|w_{k_2}^{(3)}\right|, 1\right\}.$$

Define

$$|\widetilde{w} - w| := \max\left\{\max_{k_0 \in [m_0], k_1 \in [m_1]} \left\{\left|\widetilde{w}_{k_1 k_0}^{(1)} - w_{k_1 k_0}^{(1)}\right|, \left|\widetilde{b}_{k_1}^{(1)} - b_{k_1}^{(1)}\right|\right\}, \right.$$
$$\left. \max_{k_1 \in I_{k_2}^{(2)}, k_2 \in [m_2]} \left\{\left|\widetilde{w}_{k_2 k_1}^{(2)} - w_{k_2 k_1}^{(2)}\right|, \left|\widetilde{b}_{k_2}^{(2)} - b_{k_2}^{(2)}\right|\right\}, \max_{k_2 \in [m_2]} \left|\widetilde{w}_{k_2}^{(3)} - w_{k_2}^{(3)}\right|\right\}$$

and

$$\left|\frac{\partial L_n(W)}{\partial w}\right| := \max\left\{\max_{k_0 \in [m_0], k_1 \in [m_1]} \left\{\left|\frac{\partial L_n(W)}{\partial w_{k_1 k_0}^{(1)}}\right|, \left|\frac{\partial L_n(W)}{\partial b_{k_1}^{(1)}}\right|\right\}, \right.$$
$$\left. \max_{k_1 \in I_{k_2}^{(2)}, k_2 \in [m_2]} \left\{\left|\frac{\partial L_n(W)}{\partial w_{k_2 k_1}^{(2)}}\right|, \left|\frac{\partial L_n(W)}{\partial b_{k_2}^{(2)}}\right|\right\}, \max_{k_2 \in [m_2]} \left|\frac{\partial L_n(W)}{\partial w_{k_2}^{(3)}}\right|\right\},$$

$$\left|\frac{\partial f_W}{\partial w}\right| := \max_{x \in [-B_x, B_x]^d} \left\{\max_{k_0 \in [m_0], k_1 \in [m_1]} \left\{\left|\frac{\partial f_W}{\partial w_{k_1 k_0}^{(1)}}\right|, \left|\frac{\partial f_W}{\partial b_{k_1}^{(1)}}\right|\right\}, \right.$$
$$\left. \max_{k_1 \in I_{k_2}^{(2)}, k_2 \in [m_2]} \left\{\left|\frac{\partial f_W}{\partial w_{k_2 k_1}^{(2)}}\right|, \left|\frac{\partial f_W}{\partial b_{k_2}^{(2)}}\right|\right\}, \max_{k_2 \in [m_2]} \left|\frac{\partial f_W}{\partial w_{k_2}^{(3)}}\right|\right\}.$$

## E.2   ANALYSIS OF GLOBAL CONVERGENCE OF GRADIENT DESCENT

The following three lemmas calculate Lipschitz constant of $f_W$, upper bound of $\|\nabla_W f_W\|_2$ and $\left|\frac{\partial L_n(W)}{\partial w}\right|$, Lipschitz constant of gradient of $L_n(W)$, respectively.

**Lemma E.1.** *For any $x \in [-B_x, B_x]^d$,*

$$\left| f_{\widetilde{W}}(x) - f_W(x) \right| \le \sqrt{5} L_\sigma^2 B_\sigma B_x B_3(W) B_2(W) (\widetilde{m}_2 \widetilde{m}_1 m_0)^{1/2} \left\| \widetilde{W} - W \right\|_2.$$

*Proof.* We calculate the difference of neural network functions recursively:

$$\left| f_{\widetilde{W}} - f_W \right| = \left| \widetilde{f}^{(3)} - f^{(3)} \right| \le \frac{1}{\sqrt{m_2}} \sum_{k_2=1}^{m_2} \left| \widetilde{w}_{k_2}^{(3)} \widetilde{f}_{k_2}^{(2)} - w_{k_2}^{(3)} f_{k_2}^{(2)} \right|$$

$$\le \frac{1}{\sqrt{m_2}} \sum_{k_2=1}^{m_2} \left| \widetilde{w}_{k_2}^{(3)} - w_{k_2}^{(3)} \right| \left| \widetilde{f}_{k_2}^{(2)} \right| + \frac{1}{\sqrt{m_2}} \sum_{k_2=1}^{m_2} \left| w_{k_2}^{(3)} \right| \left| \widetilde{f}_{k_2}^{(2)} - f_{k_2}^{(2)} \right|$$

$$\le \frac{B_\sigma}{\sqrt{m_2}} \sum_{k_2=1}^{m_2} \left| \widetilde{w}_{k_2}^{(3)} - w_{k_2}^{(3)} \right| + \frac{B_3(W)}{\sqrt{m_2}} \sum_{k_2=1}^{m_2} \left| \widetilde{f}_{k_2}^{(2)} - f_{k_2}^{(2)} \right|.$$

For the diffrence of the second layer,

$$\left| \widetilde{f}_{k_2}^{(2)} - f_{k_2}^{(2)} \right| \le L_\sigma \left| \widetilde{f}_{2,k_2}^{org} - f_{2,k_2}^{org} \right|$$

$$\le L_\sigma B_2(W) \sum_{k_1 \in I_{k_2}^{(2)}} \left| \widetilde{f}_{k_1}^{(1)} - f_{k_1}^{(1)} \right| + L_\sigma B_\sigma \sum_{k_1 \in I_{k_2}^{(2)}} \left| \widetilde{w}_{k_2 k_1}^{(2)} - w_{k_2 k_1}^{(2)} \right| + L_\sigma \left| \widetilde{b}_{k_2}^{(2)} - b_{k_2}^{(2)} \right|.$$

For the difference of the first layer,

$$\left| \widetilde{f}_{k_1}^{(1)} - f_{k_1}^{(1)} \right| \le L_\sigma \left| \widetilde{f}_{1,k_1}^{org} - f_{1,k_1}^{org} \right| \le L_\sigma B_x \sum_{k_0=1}^{m_0} \left| \widetilde{w}_{k_1 k_0}^{(1)} - w_{k_1 k_0}^{(1)} \right| + L_\sigma \left| \widetilde{b}_{k_1}^{(1)} - b_{k_1}^{(1)} \right|.$$

Combining all these relations yields

$$\left| f_{\widetilde{W}} - f_W \right| \le$$

$$\frac{B_\sigma}{\sqrt{m_2}} \sum_{k_2=1}^{m_2} \left| \widetilde{w}_{k_2}^{(3)} - w_{k_2}^{(3)} \right| + \frac{L_\sigma B_\sigma B_3(W)}{\sqrt{m_2}} \sum_{k_2=1}^{m_2} \sum_{k_1 \in I_{k_2}^{(2)}} \left| \widetilde{w}_{k_2 k_1}^{(2)} - w_{k_2 k_1}^{(2)} \right| + \frac{L_\sigma B_3(W)}{\sqrt{m_2}} \sum_{k_2=1}^{m_2} \left| \widetilde{b}_{k_2}^{(2)} - b_{k_2}^{(2)} \right|$$

$$+ \frac{L_\sigma^2 B_x B_3(W) B_2(W) \widetilde{m}_2}{\sqrt{m_2}} \sum_{k_1=1}^{m_1} \sum_{k_0=1}^{m_0} \left| \widetilde{w}_{k_1 k_0}^{(1)} - w_{k_1 k_0}^{(1)} \right| + \frac{L_\sigma^2 B_3(W) B_2(W) \widetilde{m}_2}{\sqrt{m_2}} \sum_{k_1=1}^{m_1} \left| \widetilde{b}_{k_1}^{(1)} - b_{k_1}^{(1)} \right|.$$

Using Hölder inquality we obtain

$$\left| f_{\widetilde{W}} - f_W \right|^2 \le 5 L_\sigma^4 B_\sigma^2 B_x^2 B_3^2(W) B_2^2(W) \widetilde{m}_2 \widetilde{m}_1 m_0 \left\| \widetilde{W} - W \right\|_2^2.$$

$\square$

**Lemma E.2.**

$$\| \nabla_W f_W \|_2 \le (2\widetilde{m}_1 + \widetilde{m}_1 m_0 + 2)^{1/2} B_\sigma B_{\sigma'}^2 B_x B_3(W) B_2(W), \quad \forall x \in [-B_x, B_x]^d,$$

$$\left| \frac{\partial L_n(W)}{\partial w} \right| \le \frac{2\widetilde{m}_2}{\sqrt{m_2}} B_\sigma B_{\sigma'}^2 B_x B_3(W) B_2(W) L_n^{1/2}(W).$$

*Proof.* We first estimate gradient of $f_W$. We calculate derivatives of $f_W$ with respect to weights in diffrent layers by definition. For the third layer,

$$\frac{\partial f_W}{\partial w_{k_2}^{(3)}} = \frac{1}{\sqrt{m_2}} f_{k_2}^{(2)}, \quad k_2 = 1, \cdots, m_2.$$

For the second layer, for $k_2 = 1, \cdots, m_2$ and $k_1 \in I_{k_2}^{(2)}$,

$$\frac{\partial f_W}{\partial w_{k_2 k_1}^{(2)}} = \frac{1}{\sqrt{m_2}} w_{k_2}^{(3)} \frac{\partial f_{k_2}^{(2)}}{\partial w_{k_2 k_1}^{(2)}} = \frac{1}{\sqrt{m_2}} w_{k_2}^{(3)} \sigma' \left( f_{2,k_2}^{org} \right) f_{k_1}^{(1)},$$

$$\frac{\partial f_W}{\partial b_{k_2}^{(2)}} = \frac{1}{\sqrt{m_2}} w_{k_2}^{(3)} \frac{\partial f_{k_2}^{(2)}}{\partial b_{k_2}^{(2)}} = \frac{1}{\sqrt{m_2}} w_{k_2}^{(3)} \sigma' \left( f_{2,k_2}^{org} \right).$$

For the first layer, for $k_1 = 1, \cdots, m_1$ and $k_0 = 1, \cdots, m_0$,

$$\frac{\partial f_W}{\partial w_{k_1 k_0}^{(1)}} = \frac{1}{\sqrt{m_2}} \sum_{k_2=1}^{m_2} w_{k_2}^{(3)} \frac{\partial f_{k_2}^{(2)}}{\partial w_{k_1 k_0}^{(1)}} = \frac{1}{\sqrt{m_2}} \sum_{k_2=1}^{m_2} w_{k_2}^{(3)} \sigma' \left( f_{2,k_2}^{org} \right) w_{k_2 k_1}^{(2)} \frac{\partial f_{k_1}^{(1)}}{\partial w_{k_1 k_0}^{(1)}}$$

$$= \frac{1}{\sqrt{m_2}} \sum_{k_2 \in I_{k_1}^{(1)}} w_{k_2}^{(3)} \sigma' \left( f_{2,k_2}^{org} \right) w_{k_2 k_1}^{(2)} \frac{\partial f_{k_1}^{(1)}}{\partial w_{k_1 k_0}^{(1)}} = \frac{1}{\sqrt{m_2}} \sum_{k_2 \in I_{k_1}^{(1)}} w_{k_2}^{(3)} \sigma' \left( f_{2,k_2}^{org} \right) w_{k_2 k_1}^{(2)} \sigma' \left( f_{1,k_1}^{org} \right) f_{k_0}^{(0)}.$$

Similarly,

$$\frac{\partial f_W}{\partial b_{k_1}^{(1)}} = \frac{1}{\sqrt{m_2}} \sum_{k_2=1}^{m_2} w_{k_2}^{(3)} \sigma' \left( f_{2,k_2}^{org} \right) w_{k_2 k_1}^{(2)} \sigma' \left( f_{1,k_1}^{org} \right) = \frac{1}{\sqrt{m_2}} \sum_{k_2 \in I_{k_1}^{(1)}} w_{k_2}^{(3)} \sigma' \left( f_{2,k_2}^{org} \right) w_{k_2 k_1}^{(2)} \sigma' \left( f_{1,k_1}^{org} \right).$$

Combining all these upper bounds for partial derivatives we obtain an upper bound for the gradient of $f_W$:

$$\|\nabla_W f_W\|_2^2$$

$$= \sum_{k_1=1}^{m_1} \sum_{k_0=1}^{m_0} \left| \frac{\partial f_W}{\partial w_{k_1 k_0}^{(1)}} \right|^2 + \sum_{k_2=1}^{m_2} \sum_{k_1 \in I_{k_2}^{(2)}} \left| \frac{\partial f_W}{\partial w_{k_2 k_1}^{(2)}} \right|^2 + \sum_{k_2=1}^{m_2} \left| \frac{\partial f_W}{\partial w_{k_2}^{(3)}} \right|^2 + \sum_{k_1=1}^{m_1} \left| \frac{\partial f_W}{\partial b_{k_1}^{(1)}} \right|^2 + \sum_{k_2=1}^{m_2} \left| \frac{\partial f_W}{\partial b_{k_2}^{(2)}} \right|^2$$

$$\leq (2\widetilde{m}_1 + \widetilde{m}_1 m_0 + 2) B_\sigma^2 B_{\sigma'}^4 B_x^2 B_3^2(W) B_2^2(W)$$

and

$$\left| \frac{\partial L_n(W)}{\partial w} \right|^2 \leq 4 L_n(W) \left| \frac{\partial f_W}{\partial w} \right|^2 \leq \frac{4\widetilde{m}_2^2}{m_2} B_\sigma^2 B_{\sigma'}^4 B_x^2 B_3^2(W) B_2^2(W) L_n(W).$$

$\square$

**Lemma E.3.**

$$\left\| \nabla_{\widetilde{W}} L_n \left( \widetilde{W} \right) - \nabla_W L_n \left( W \right) \right\|_2 \leq 2 \left[ \sqrt{\frac{86}{m_2}} \widetilde{m}_2 \widetilde{m}_1 m_0 L_n^{1/2}(W) + \sqrt{5 \widetilde{m}_2 \widetilde{m}_1 m_0 (2\widetilde{m}_1 + \widetilde{m}_1 m_0 + 2)} \right]$$

$$L_\sigma^2 L_{\sigma'} B_\sigma^2 B_{\sigma'}^2 B_x^2 B_3(W) B_3(\widetilde{W}) \max \left\{ B_2^2(\widetilde{W}), B_2^2(W) \right\} \left\| \widetilde{W} - W \right\|_2.$$

*Proof.* We calculate difference of partial derivatives layer by layer. For the third layer,

$$\left| \frac{\partial f_{\widetilde{W}}}{\partial \widetilde{w}_{k_2}^{(3)}} - \frac{\partial f_W}{\partial w_{k_2}^{(3)}} \right| = \frac{1}{\sqrt{m_2}} \left| \widetilde{f}_{k_2}^{(2)} - f_{k_2}^{(2)} \right| \leq \frac{L_\sigma}{\sqrt{m_2}} \left| \widetilde{f}_{2,k_2}^{org} - f_{2,k_2}^{org} \right|.$$

For the second layer,

$$\left| \frac{\partial f_{\widetilde{W}}}{\partial \widetilde{w}_{k_2 k_1}^{(2)}} - \frac{\partial f_W}{\partial w_{k_2 k_1}^{(2)}} \right| = \frac{1}{\sqrt{m_2}} \left| \widetilde{w}_{k_2}^{(3)} \sigma' \left( \widetilde{f}_{2,k_2}^{org} \right) \widetilde{f}_{k_1}^{(1)} - w_{k_2}^{(3)} \sigma' \left( f_{2,k_2}^{org} \right) f_{k_1}^{(1)} \right|$$

$$\leq \frac{1}{\sqrt{m_2}} B_\sigma B_{\sigma'} \left| \widetilde{w}_{k_2}^{(3)} - w_{k_2}^{(3)} \right| + \frac{1}{\sqrt{m_2}} B_\sigma B_3(W) \left| \sigma' \left( \widetilde{f}_{2,k_2}^{org} \right) - \sigma' \left( f_{2,k_2}^{org} \right) \right| + \frac{1}{\sqrt{m_2}} B_{\sigma'} B_3(W) \left| \widetilde{f}_{k_1}^{(1)} - f_{k_1}^{(1)} \right|$$

$$\leq \frac{1}{\sqrt{m_2}} B_\sigma B_{\sigma'} \left| \widetilde{w}_{k_2}^{(3)} - w_{k_2}^{(3)} \right| + \frac{1}{\sqrt{m_2}} L_{\sigma'} B_\sigma B_3(W) \left| \widetilde{f}_{2,k_2}^{org} - f_{2,k_2}^{org} \right| + \frac{1}{\sqrt{m_2}} L_\sigma B_{\sigma'} B_3(W) \left| \widetilde{f}_{1,k_1}^{org} - f_{1,k_1}^{org} \right|.$$

For the first layer,

$$\left| \frac{\partial f_{\widetilde{W}}}{\partial \widetilde{w}_{k_1 k_0}^{(1)}} - \frac{\partial f_W}{\partial w_{k_1 k_0}^{(1)}} \right|$$

$$= \frac{1}{\sqrt{m_2}} \left| \sum_{k_2 \in I_{k_1}^{(1)}} \widetilde{w}_{k_2}^{(3)} \sigma' \left( \widetilde{f}_{2,k_2}^{org} \right) \widetilde{w}_{k_2 k_1}^{(2)} \sigma' \left( \widetilde{f}_{1,k_1}^{org} \right) x_{k_0} - \sum_{k_2 \in I_{k_1}^{(1)}} w_{k_2}^{(3)} \sigma' \left( f_{2,k_2}^{org} \right) w_{k_2 k_1}^{(2)} \sigma' \left( f_{1,k_1}^{org} \right) x_{k_0} \right|$$

$$\leq \frac{1}{\sqrt{m_2}} B_{\sigma'}^2 B_x B_2(\widetilde{W}) \sum_{k_2 \in I_{k_1}^{(1)}} \left| \widetilde{w}_{k_2}^{(3)} - w_{k_2}^{(3)} \right| + \frac{1}{\sqrt{m_2}} B_{\sigma'} B_x B_3(W) B_2(\widetilde{W}) \sum_{k_2 \in I_{k_1}^{(1)}} \left| \sigma' \left( \widetilde{f}_{2,k_2}^{org} \right) - \sigma' \left( f_{2,k_2}^{org} \right) \right|$$

$$+ \frac{1}{\sqrt{m_2}} B_{\sigma'}^2 B_x B_3(W) \sum_{k_2 \in I_{k_1}^{(1)}} \left| \widetilde{w}_{k_2 k_1}^{(2)} - w_{k_2 k_1}^{(2)} \right| + \frac{1}{\sqrt{m_2}} \widetilde{m}_2 B_{\sigma'} B_x B_3(W) B_2(W) \left| \sigma' \left( \widetilde{f}_{1,k_1}^{org} \right) - \sigma' \left( f_{1,k_1}^{org} \right) \right|$$

$$\leq \frac{1}{\sqrt{m_2}} B_{\sigma'}^2 B_x B_2(\widetilde{W}) \sum_{k_2 \in I_{k_1}^{(1)}} \left| \widetilde{w}_{k_2}^{(3)} - w_{k_2}^{(3)} \right| + \frac{1}{\sqrt{m_2}} L_{\sigma'} B_{\sigma'} B_x B_3(W) B_2(\widetilde{W}) \sum_{k_2 \in I_{k_1}^{(1)}} \left| \widetilde{f}_{2,k_2}^{org} - f_{2,k_2}^{org} \right|$$

$$+ \frac{1}{\sqrt{m_2}} B_{\sigma'}^2 B_x B_3(W) \sum_{k_2 \in I_{k_1}^{(1)}} \left| \widetilde{w}_{k_2 k_1}^{(2)} - w_{k_2 k_1}^{(2)} \right| + \frac{1}{\sqrt{m_2}} \widetilde{m}_2 L_{\sigma'} B_{\sigma'} B_x B_3(W) B_2(W) \left| \widetilde{f}_{1,k_1}^{org} - f_{1,k_1}^{org} \right|.$$

Similarly we can derive upper bounds with respect to the bias.

$$\left| \frac{\partial f_{\widetilde{W}}}{\partial \widetilde{b}_{k_2}^{(2)}} - \frac{\partial f_W}{\partial b_{k_2}^{(2)}} \right| \leq \frac{1}{\sqrt{m_2}} B_{\sigma'} \left| \widetilde{w}_{k_2}^{(3)} - w_{k_2}^{(3)} \right| + \frac{1}{\sqrt{m_2}} L_{\sigma'} B_3(W) \left| \widetilde{f}_{2,k_2}^{org} - f_{2,k_2}^{org} \right|,$$

$$\left| \frac{\partial f_{\widetilde{W}}}{\partial \widetilde{b}_{k_1}^{(1)}} - \frac{\partial f_W}{\partial b_{k_1}^{(1)}} \right| \leq$$

$$\frac{1}{\sqrt{m_2}} B_{\sigma'}^2 B_2(\widetilde{W}) \sum_{k_2 \in I_{k_1}^{(1)}} \left| \widetilde{w}_{k_2}^{(3)} - w_{k_2}^{(3)} \right| + \frac{1}{\sqrt{m_2}} L_{\sigma'} B_{\sigma'} B_3(W) B_2(\widetilde{W}) \sum_{k_2 \in I_{k_1}^{(1)}} \left| \widetilde{f}_{2,k_2}^{org} - f_{2,k_2}^{org} \right|$$

$$+ \frac{1}{\sqrt{m_2}} B_{\sigma'}^2 B_3(W) \sum_{k_2 \in I_{k_1}^{(1)}} \left| \widetilde{w}_{k_2 k_1}^{(2)} - w_{k_2 k_1}^{(2)} \right| + \frac{1}{\sqrt{m_2}} \widetilde{m}_2 L_{\sigma'} B_{\sigma'} B_3(W) B_2(W) \left| \widetilde{f}_{1,k_1}^{org} - f_{1,k_1}^{org} \right|.$$

Next we derive upper bounds for $\left| \widetilde{f}_{1,k_1}^{org} - f_{1,k_1}^{org} \right|$ and $\left| \widetilde{f}_{2,k_2}^{org} - f_{2,k_2}^{org} \right|$ which arise from above upper bounds.

$$\left| \widetilde{f}_{1,k_1}^{org} - f_{1,k_1}^{org} \right| \leq \sum_{k_0=1}^{m_0} \left| \widetilde{w}_{k_1 k_0}^{(1)} f_{k_0}^{(0)} - w_{k_1 k_0}^{(1)} f_{k_0}^{(0)} \right| + \left| \widetilde{b}_{k_1}^{(1)} - b_{k_1}^{(1)} \right|$$

$$\leq \sum_{k_0=1}^{m_0} \left| \widetilde{w}_{k_1 k_0}^{(1)} - w_{k_1 k_0}^{(1)} \right| \left| \widetilde{f}_{k_0}^{(0)} \right| + \left| \widetilde{b}_{k_1}^{(1)} - b_{k_1}^{(1)} \right| \leq B_x \sum_{k_0=1}^{m_0} \left| \widetilde{w}_{k_1 k_0}^{(1)} - w_{k_1 k_0}^{(1)} \right| + \left| \widetilde{b}_{k_1}^{(1)} - b_{k_1}^{(1)} \right|.$$

$$\left| \widetilde{f}_{2,k_2}^{org} - f_{2,k_2}^{org} \right| \leq \sum_{k_1 \in I_{k_2}^{(2)}} \left| \widetilde{w}_{k_2 k_1}^{(2)} \widetilde{f}_{k_1}^{(1)} - w_{k_2 k_1}^{(2)} f_{k_1}^{(1)} \right| + \left| \widetilde{b}_{k_2}^{(2)} - b_{k_2}^{(2)} \right|$$

$$\leq \sum_{k_1 \in I_{k_2}^{(2)}} \left| \widetilde{w}_{k_2 k_1}^{(2)} - w_{k_2 k_1}^{(2)} \right| \left| \widetilde{f}_{k_1}^{(1)} \right| + \sum_{k_1 \in I_{k_2}^{(2)}} \left| w_{k_2 k_1}^{(2)} \right| \left| \widetilde{f}_{k_1}^{(1)} - f_{k_1}^{(1)} \right| + \left| \widetilde{b}_{k_2}^{(2)} - b_{k_2}^{(2)} \right|$$

$$\leq B_{\sigma} \sum_{k_1 \in I_{k_2}^{(2)}} \left| \widetilde{w}_{k_2 k_1}^{(2)} - w_{k_2 k_1}^{(2)} \right| + B_2(W) \sum_{k_1 \in I_{k_2}^{(2)}} \left| \widetilde{f}_{k_1}^{(1)} - f_{k_1}^{(1)} \right| + \left| \widetilde{b}_{k_2}^{(2)} - b_{k_2}^{(2)} \right|$$

$$\leq B_{\sigma} \sum_{k_1 \in I_{k_2}^{(2)}} \left| \widetilde{w}_{k_2 k_1}^{(2)} - w_{k_2 k_1}^{(2)} \right| + L_{\sigma} B_2(W) \sum_{k_1 \in I_{k_2}^{(2)}} \left| \widetilde{f}_{1,k_1}^{org} - f_{1,k_1}^{org} \right| + \left| \widetilde{b}_{k_2}^{(2)} - b_{k_2}^{(2)} \right|$$

$$\leq B_{\sigma} \sum_{k_1 \in I_{k_2}^{(2)}} \left| \widetilde{w}_{k_2 k_1}^{(2)} - w_{k_2 k_1}^{(2)} \right| + L_{\sigma} B_x B_2(W) \sum_{k_1 \in I_{k_2}^{(2)}} \sum_{k_0=1}^{m_0} \left| \widetilde{w}_{k_1 k_0}^{(1)} - w_{k_1 k_0}^{(1)} \right|$$

$$+ L_\sigma B_2(W) \sum_{k_1 \in I_{k_2}^{(2)}} \left| \widetilde{b}_{k_1}^{(1)} - b_{k_1}^{(1)} \right| + \left| \widetilde{b}_{k_2}^{(2)} - b_{k_2}^{(2)} \right|.$$

Here in the final step we use the upper bound for $\left| \widetilde{f}_{1,k_1}^{org} - f_{1,k_1}^{org} \right|$.

Combining all these inequalities we obtain

$$\left\| \nabla_{\widetilde{W}} f_{\widetilde{W}} - \nabla_W f_W \right\|_2^2$$

$$= \sum_{k_1=1}^{m_1} \sum_{k_0=1}^{m_0} \left| \frac{\partial f_{\widetilde{W}}}{\partial \widetilde{w}_{k_1 k_0}^{(1)}} - \frac{\partial f_W}{\partial w_{k_1 k_0}^{(1)}} \right|^2 + \sum_{k_2=1}^{m_2} \sum_{k_1 \in I_{2,k_2}} \left| \frac{\partial f_{\widetilde{W}}}{\partial \widetilde{w}_{k_2 k_1}^{(2)}} - \frac{\partial f_W}{\partial w_{k_2 k_1}^{(2)}} \right|^2 + \sum_{k_2=1}^{m_2} \left| \frac{\partial f_{\widetilde{W}}}{\partial \widetilde{w}_{k_2}^{(3)}} - \frac{\partial f_W}{\partial w_{k_2}^{(3)}} \right|^2$$

$$+ \sum_{k_1=1}^{m_1} \left| \frac{\partial f_{\widetilde{W}}}{\partial \widetilde{b}_{k_1}^{(1)}} - \frac{\partial f_W}{\partial b_{k_1}^{(1)}} \right|^2 + \sum_{k_2=1}^{m_2} \left| \frac{\partial f_{\widetilde{W}}}{\partial \widetilde{b}_{k_2}^{(2)}} - \frac{\partial f_W}{\partial b_{k_2}^{(2)}} \right|^2$$

$$\leq \frac{86}{m_2} L_\sigma^4 L_{\sigma'}^2 B_\sigma^2 B_{\sigma'}^2 B_x^4 B_3^2(W) \max\{B_2^4(W), B_2^4(\widetilde{W})\} \widetilde{m}_2^2 \widetilde{m}_1^2 m_0^2 \|\widetilde{W} - W\|_2^2. \tag{E.1}$$

Combining Lemma E.1, Lemma E.2 and the above upper bound we derive that

$$\left\| \nabla_{\widetilde{W}} L_n \left( \widetilde{W} \right) - \nabla_W L_n \left( W \right) \right\|_2$$

$$\leq \frac{2}{n} \sum_{i=1}^n |f_W(X_i) - Y_i| \left\| \nabla_{\widetilde{W}} f_{\widetilde{W}}(X_i) - \nabla_W f_W(X_i) \right\|_2 + \frac{2}{n} \sum_{i=1}^n \left| f_{\widetilde{W}}(X_i) - f_W(X_i) \right| \left\| \nabla_{\widetilde{W}} f_{\widetilde{W}}(X_i) \right\|_2$$

$$\leq 2 \left[ \sqrt{\frac{86}{m_2}} \widetilde{m}_2 \widetilde{m}_1 m_0 L_n^{1/2}(W) + \sqrt{5 \widetilde{m}_2 \widetilde{m}_1 m_0 (2\widetilde{m}_1 + \widetilde{m}_1 m_0 + 2)} \right]$$

$$L_\sigma^2 L_{\sigma'} B_\sigma^2 B_{\sigma'}^2 B_x^2 B_3(W) B_3(\widetilde{W}) \max \left\{ B_2^2(\widetilde{W}), B_2^2(W) \right\} \left\| \widetilde{W} - W \right\|_2.$$

$$\square$$

The following matrix Chernoff bound can be found in Tropp (2012).

**Proposition E.1** (matrix Chernoff bound). *Assume that $A_k \in \mathbb{R}^{n \times n}(k \in [m])$ are positive semi-definite matrices and $\lambda_{\max}(A_k) \leq B(k \in [m])$. For $\delta \in [0, 1)$,*

$$\mathbb{P}\left( \lambda_{\min}\left( \sum_{k=1}^m A_k \right) \leq (1 - \delta)\lambda_{\min}\left( \mathbb{E}\sum_{k=1}^m A_k \right) \right) \leq n \left[ \frac{e^{-\delta}}{(1-\delta)^{1-\delta}} \right]^{\lambda_{\min}\left( \mathbb{E}\sum_{k=1}^m A_k \right)/B}.$$

PL condition is a sufficient condition for global convergence of optimization algorithm. The next lemma, which is based on the analysis of Theorem 2 in Liu et al. (2022), says that as long as the width of a neural network(reflected by $m_2$ in this paper) is sufficiently large, $L_n(W)$ satisfies the PL condition.

**Lemma E.4.** *Initialize $W_0$ by (5). Let $t_2, t_3, t_4 > 0$. Let*

$$B_3^{(t_2)} = C^{1/2} \sigma_3 \ln^{1/2} \frac{m_2}{t_2}, \quad B_2^{(t_3)} = C^{1/2} \sigma_2 \ln^{1/2} \frac{m_2 \widetilde{m}_1}{t_3}.$$

*Assume $|w - w_0| \leq \min \left\{ B_2^{(t_3)}, B_3^{(t_2)} \right\}$. If*

$$m_2 \geq \max \left\{ m_{2,1}^{(low)}, m_{2,2}^{(low)} \right\}$$

*with*

$$m_{2,1}^{(low)} = \frac{2n B_{\sigma'}^2 B_\sigma^2 \left( B_3^{(t_6)} \right)^2 \log_{e/2} \frac{n}{t_4}}{C_3 \widetilde{m}_1 \sigma_3^2 \lambda_{\min}(K_0)},$$

$$m_{2,2}^{(low)} = \frac{1409024n^2(2\widetilde{m}_1 + \widetilde{m}_1 m_0 + 2)\widetilde{m}_2^2 \widetilde{m}_1^2 m_0^2 B_x^6 L_\sigma^4 L_{\sigma'}^2 B_\sigma^4 B_{\sigma'}^6 \left(\max\left\{B_3^{(t_2)}, B_2^{(t_3)}\right\}\right)^{12}}{C_3^2 \widetilde{m}_1^2 \sigma_3^4 \lambda_{\min}^2(K_0)},$$

*where*

$$C_3 = 2 \int_0^\infty \left[1 - 2e^{-t^2/\left(C\left(\widetilde{m}_1 B_\sigma^2 + 1\right)\sigma_2^2\right)}\right] \sigma'(t)\sigma''(t)dt,$$

*then with probability $1 - t_2 - t_3 - t_4$ over $W_0$,*

$$\|\nabla_{W^{(2)}} L_n(W)\|_2^2 \geq \mu L_n(W)$$

*with*

$$\mu = C_3 \widetilde{m}_1 \sigma_3^2 \lambda_{\min}(K_0).$$

*Proof.* We can first give a lower bound of the gradient of empirical loss in terms of the minimum eigenvalue of NTK matrix. To make derivation clear, we introduce some notations. Denote

$$D_{W^{(2)}} F = [\nabla_{W^{(2)}} f_W(X_1), \cdots, \nabla_{W^{(2)}} f_W(X_n)]$$

and

$$F_W(X) - Y = [f_W(X_1) - Y_1, \cdots, f_W(X_n) - Y_n]^T,$$

then

$$\nabla_{W^{(2)}} L_n(W) = \frac{2}{n}\sum_{i=1}^n [f_W(X_i) - Y_i]\nabla_{W^{(2)}} f_W(X_i) = \frac{2}{n}D_{W^{(2)}} F[F_W(X) - Y].$$

With these notations, we have

$$\|\nabla_{W^{(2)}} L_n(W)\|_2^2 = (\nabla_{W^{(2)}} L_n(W))^T \nabla_{W^{(2)}} L_n(W)$$

$$= \frac{4}{n^2}[F_W(X) - Y]^T (D_{W^{(2)}} F)^T D_{W^{(2)}} F[F_W(X) - Y] = \frac{4}{n^2}[F_W(X) - Y]^T K_{W^{(2)}}(W)[F_W(X) - Y]$$

$$\geq \frac{4}{n^2}\lambda_{\min}(K_{W^{(2)}}(W))\|F_W(X) - Y\|_2^2 = 4\lambda_{\min}(K_{W^{(2)}}(W))L_n(W), \tag{E.2}$$

where $K_{W^{(2)}}(W) := (D_{W^{(2)}} F)^T D_{W^{(2)}} F$ and

$$(K_{W^{(2)}}(W))_{ij} = \frac{1}{m_2}\sum_{k_2=1}^{m_2}\sum_{k_1 \in I_{k_1}^{(1)}} \left(w_{k_2}^{(3)}\right)^2 \sigma'\left(f_{2,k_2,i}^{org}\right) f_{k_1,i}^{(1)} \sigma'\left(f_{2,k_2,j}^{org}\right) f_{k_1,j}^{(1)}.$$

It can be checked that we can rewrite $K_{W^{(2)}}$ in the following form:

$$K_{W^{(2)}} = \frac{1}{m_2}\sum_{k_2=1}^{m_2}\sum_{k_1 \in I_{k_1}^{(1)}} \left(w_{k_2}^{(3)}\right)^2 \Lambda_{k_2} K_{k_1}^{inn} \Lambda_{k_2}, \tag{E.3}$$

where

$$(\Lambda_{k_2})_{ij} = \begin{cases} \sigma'\left(f_{2,k_2,i}^{org}\right), & i = j \\ 0, & i \neq j \end{cases}, \quad i, j = 1, \cdots, n$$

and

$$\left(K_{k_1}^{inn}\right)_{ij} = f_{k_1,i}^{(1)} f_{k_1,j}^{(1)}, \quad i, j = 1, \cdots, n.$$

If we denote $F_{k_1}^{(1)} = \left(f_{k_1,1}^{(1)}, f_{k_1,2}^{(1)}, \cdots, f_{k_1,n}^{(1)}\right)$, then $K_{k_1}^{inn} = F_{k_1}^{(1)T} F_{k_1}^{(1)}$.

We next give a lower bound for $\lambda_{\min}(K_{W^{(2)}}(W))$. Decompose $K_{W^{(2)}}(W_0)$ into two parts: $K_{W^{(2)}}(W_0) = [K_{W^{(2)}}(W_0) - K_{W^{(2)}}(W)] + K_{W^{(2)}}(W)$. Applying Weyl's inequality($K_{W^{(2)}}$ is symmetric) yields

$$\lambda_{\min}(K_{W^{(2)}}(W_0)) \leq \|K_{W^{(2)}}(W_0) - K_{W^{(2)}}(W)\|_2 + \lambda_{\min}(K_{W^{(2)}}(W)).$$

In other words,
$$\lambda_{\min}(K_{W^{(2)}}(W)) \geq \lambda_{\min}(K_{W^{(2)}}(W_0)) - \|K_{W^{(2)}}(W) - K_{W^{(2)}}(W_0)\|_2. \tag{E.4}$$

For the difference of NTK matrix, using Lemma E.2 and (E.1), we have
$$\|K_{W^{(2)}}(W) - K_{W^{(2)}}(W_0)\|_2 = \|D_{W^{(2)}}F(W)^T D_{W^{(2)}}F(W) - D_{W^{(2)}}F(W_0)^T D_{W^{(2)}}F(W_0)\|_2$$
$$\leq \|D_{W^{(2)}}F(W) - D_{W^{(2)}}F(W_0)\|_2 \|D_{W^{(2)}}F(W)\|_2 + \|D_{W^{(2)}}F(W_0)\|_2 \|D_{W^{(2)}}F(W) - D_{W^{(2)}}F(W_0)\|_2$$
$$\leq 2\sqrt{\frac{86}{m_2}} n(2\widetilde{m}_1 + \widetilde{m}_1 m_0 + 2)^{1/2} \widetilde{m}_2 \widetilde{m}_1 m_0 B_x^3 L_\sigma^2 L_{\sigma'} B_\sigma^2 B_{\sigma'}^3$$
$$\max\{B_3^2(W), B_3^2(W_0)\} \max\{B_2^3(W), B_2^3(W_0)\} \|W - W_0\|_2.$$

By Proposition 2.5.2 in Vershynin (2018) and union bound we have with probability at least $1 - t_2 - t_3$ over $W_0$,
$$B_3(W_0) = B_3^{(t_2)}, B_2(W_0) = B_2^{(t_3)}. \tag{E.5}$$

Then by assumption $|w - w_0| \leq \min\left\{B_2^{(t_3)}, B_3^{(t_2)}\right\}$, we have with probability at least $1 - t_2 - t_3$ over $W_0$,
$$B_3(W) = \max\left\{\max_{k_2}\left|w_{k_2}^{(3)}\right|, 1\right\} \leq \max\left\{\max_{k_2}\left|w_{k_2}^{(3)} - (w_0)_{k_2}^{(3)}\right| + \left|(w_0)_{k_2}^{(3)}\right|, 1\right\}$$
$$\leq \max\left\{2B_3^{(t_2)}, 1\right\} = 2B_3^{(t_2)}$$

and similarly $B_2(W) \leq 2B_2^{(t_3)}$. Hence we conclude that with probability at least $1 - t_2 - t_3$ over $W_0$,
$$\|K_{W^{(2)}}(W) - K_{W^{(2)}}(W_0)\|_2$$
$$\leq 64\sqrt{\frac{86}{m_2}} n(2\widetilde{m}_1 + \widetilde{m}_1 m_0 + 2)^{1/2} \widetilde{m}_2 \widetilde{m}_1 m_0 B_x^3 L_\sigma^2 L_{\sigma'} B_\sigma^2 B_{\sigma'}^3 \left(B_3^{(t_2)}\right)^2 \left(B_2^{(t_3)}\right)^3 \min\left\{B_3^{(t_2)}, B_2^{(t_3)}\right\}. \tag{E.6}$$

We next study $\lambda_{\min}(K_{W^{(2)}}(W_0))$. Applying Proposition E.1 with $\delta = \frac{1}{2}$ and noting that with probability at least $1 - t_2$ over $W_0$,

$$\lambda_{\max}\left(\frac{1}{m_2}\left((w_0)_{k_2}^{(3)}\right)^2 \Lambda_{k_2}(W_0) K_{k_1}^{inn}(W_0) \Lambda_{k_2}(W_0)\right) \leq \frac{\left((w_0)_{k_2}^{(3)}\right)^2 \|\Lambda_{k_2}(W_0)\|_2^2 \|K_{k_1}^{inn}(W_0)\|_2}{m_2}$$
$$\leq \frac{B_3^2(W_0) B_{\sigma'}^2 \left\|F_{k_1}^{(1)}\right\|_2^2}{m_2} \leq \frac{n B_3^2(W_0) B_{\sigma'}^2 B_\sigma^2}{m_2} = \frac{n\left(B_3^{(t_2)}\right)^2 B_{\sigma'}^2 B_\sigma^2}{m_2},$$

we have with probability at least $1 - t_2 - n\left(\frac{2}{e}\right)^{\frac{m_2 \lambda_{\min}(\mathbb{E}_{W_0} K_{W^{(2)}}(W_0))}{2n\left(B_3^{(t_2)}\right)^2 B_{\sigma'}^2 B_\sigma^2}}$ over $W_0$,
$$\lambda_{\min}\left(K_{W^{(2)}}(W_0)\right) \geq \frac{1}{2}\lambda_{\min}\left(\mathbb{E}_{W_0} K_{W^{(2)}}(W_0)\right). \tag{E.7}$$

Using (E.3) and Weyl's inequality(all $\Lambda_{k_2}(W_0)$ and $K_{k_1}^{inn}(W_0)$ are symmetric), we have

$$\lambda_{\min}\left(\mathbb{E}_{W_0} K_{W^{(2)}}(W_0)\right) = \lambda_{\min}\left(\frac{1}{m_2}\sum_{k_2=1}^{m_2}\sum_{k_1 \in I_{k_1}^{(1)}} \mathbb{E}_{W_0}\left((w_0)_{k_2}^{(3)}\right)^2 \mathbb{E}_{W_0}\Lambda_{k_2}(W_0) K_{k_1}^{inn}(W_0)\Lambda_{k_2}(W_0)\right)$$
$$\geq \frac{1}{m_2}\sum_{k_2=1}^{m_2}\sum_{k_1 \in I_{k_1}^{(1)}} \mathbb{E}_{W_0}\left((w_0)_{k_2}^{(3)}\right)^2 \lambda_{\min}\left(\mathbb{E}_{W_0}\Lambda_{k_2}(W_0) K_{k_1}^{inn}(W_0)\Lambda_{k_2}(W_0)\right)$$
$$\geq \frac{\sigma_3^2}{m_2}\sum_{k_2=1}^{m_2}\sum_{k_1 \in I_{k_1}^{(1)}} \lambda_{\min}\left(\mathbb{E}_{W_0}\Lambda_{k_2}(W_0) K_{k_1}^{inn}(W_0)\Lambda_{k_2}(W_0)\right). \tag{E.8}$$

For the minimal eigenvalue of $\mathbb{E}_{W_0} \Lambda_{k_2}(W_0) K_{k_1}^{inn}(W_0) \Lambda_{k_2}(W_0)$,

$$\lambda_{\min}\left(\mathbb{E}_{W_0} \Lambda_{k_2}(W_0) K_{k_1}^{inn}(W_0) \Lambda_{k_2}(W_0)\right) = \lambda_{\min}\left(\mathbb{E}_{W_0} \Lambda_{k_2}(W_0)(\mathbb{E}_{W_0} K_{k_1}^{inn}(W_0)) \Lambda_{k_2}(W_0))\right]$$

$$= \min_{v \in \mathbb{R}^n} \frac{\mathbb{E}_{W_0} v^T \Lambda_{k_2}(W_0)(\mathbb{E}_{W_0} K_{k_1}^{inn}(W_0)) \Lambda_{k_2}(W_0) v}{v^T v} = \min_{v \in \mathbb{R}^n} \frac{\mathbb{E}_{W_0}(\Lambda_{k_2}(W_0) v)^T (\mathbb{E}_{W_0} K_{k_1}^{inn}(W_0)) \Lambda_{k_2}(W_0) v}{v^T v}$$

$$\geq \lambda_{\min}(\mathbb{E}_{W_0} K_{k_1}^{inn}(W_0)) \min_{v \in \mathbb{R}^n} \frac{\mathbb{E}_{W_0}(\Lambda_{k_2}(W_0) v)^T \Lambda_{k_2}(W_0) v}{v^T v}. \tag{E.9}$$

To evaluate the minimum, we need to estimate $f_{2,k_2,i}^{org}(W_0)$. Applying Proposition 2.6.1 in Vershynin (2018), we have for $i \in [n]$,

$$\left\| f_{2,k_2,i}^{org}(W_0) \right\|_{\psi_2}^2 = \left\| \sum_{k_1 \in I_{k_2}^{(2)}} (w_0)_{k_2 k_1}^{(2)} \sigma\left(f_{1,k_1,i}^{org}(W_0)\right) + (b_0)_{k_2}^{(2)} \right\|_{\psi_2}^2$$

$$\leq C \sum_{k_1 \in I_{k_2}^{(2)}} \left\| (w_0)_{k_2 k_1}^{(2)} \sigma\left(f_{1,k_1,i}^{org}(W_0)\right) \right\|_{\psi_2}^2 + C \left\| (b_0)_{k_2}^{(2)} \right\|_{\psi_2}^2 \leq C \left(\widetilde{m}_1 B_\sigma^2 + 1\right) \sigma_2^2.$$

Hence by Proposition 2.5.2 in Vershynin (2018), $\left| f_{2,k_2,i}^{org}(W_0) \right| \leq t$ with probability at least $1 - 2e^{-t^2/(C(\widetilde{m}_1 B_\sigma^2 + 1)\sigma_2^2)}$ over $W_0$. Let $\tau = \sigma'(t)^2$, by substitution rule and the monotonicity of $\sigma'$ we derive that

$$\mathbb{E}_{W_0} \sigma'\left(f_{2,k_2,i}^{org}(W_0)\right)^2 = \int_0^{\frac{1}{16}} \mathbb{P}\left(\sigma'\left(f_{2,k_2,i}^{org}(W_0)\right)^2 \geq \tau\right) d\tau$$

$$= \int_0^\infty \mathbb{P}\left(\sigma'\left(f_{2,k_2,i}^{org}(W_0)\right)^2 \geq \sigma'(t)^2\right) 2\sigma'(t)\sigma''(t) dt = 2 \int_0^\infty \mathbb{P}\left(\left| f_{2,k_2,i}^{org}(W_0) \right| \leq t\right) \sigma'(t)\sigma''(t) dt$$

$$\geq 2 \int_0^\infty \left[1 - 2e^{-t^2/(C(\widetilde{m}_1 B_\sigma^2 + 1)\sigma_2^2)}\right] \sigma'(t)\sigma''(t) dt = C_3.$$

Therefore

$$\mathbb{E}_{W_0} \|\Lambda_{k_2}(W_0) v\|_2^2 = \mathbb{E}_{W_0} \sum_{i=1}^n \sigma'\left(f_{2,k_2,i}^{org}(W_0)\right)^2 v_i^2$$

$$= \sum_{i=1}^n \left[\mathbb{E}_{W_0} \sigma'\left(f_{2,k_2,i}^{org}(W_0)\right)^2\right] v_i^2 \geq C_3 \|v\|_2^2. \tag{E.10}$$

Combining (E.8)-(E.10) we obtain

$$\lambda_{\min}\left(\mathbb{E}_{W_0} K_{W^{(2)}}(W_0)\right) \geq \frac{C_3 \sigma_3^2}{m_2} \sum_{k_2=1}^{m_2} \sum_{k_1 \in I_{k_2}^{(2)}} \lambda_{\min}(\mathbb{E}_{W_0} K_{k_1}^{inn}(W_0)) = C_3 \widetilde{m}_1 \sigma_3^2 \lambda_{\min}(K_0).$$

Using (E.7) we obtain that with probability at least $1 - t_2 - n\left(\frac{2}{e}\right)^{\frac{C_3 \widetilde{m}_1 m_2 \mu_r^2(\sigma) \sigma_3^2}{2n(d+1)^r \left(B_3^{(t_2)}\right)^2 B_{\sigma'}^2, B_\sigma^2}}$ over $W_0$,

$$\lambda_{\min}\left(K_{W^{(2)}}(W_0)\right) \geq \frac{1}{2} \lambda_{\min}\left(\mathbb{E}_{W_0} K_{W^{(2)}}(W_0)\right) \geq \frac{1}{2} C_3 \widetilde{m}_1 \sigma_3^2 \lambda_{\min}(K_0). \tag{E.11}$$

Combining (E.4), (E.6) and (E.11) we conclude that with probability at least $1 - t_2 - t_3 - t_4$ over $W_0$,

$$\lambda_{\min}\left(K_{W^{(2)}}(W)\right) \geq \frac{1}{4} C_3 \widetilde{m}_1 \sigma_3^2 \lambda_{\min}(K_0) \tag{E.12}$$

provided $m_2 \geq \max\left\{m_{2,1}^{(low)}, m_{2,2}^{(low)}\right\}$. (E.12) together with (E.2) implies the PL inequality constant.

$\square$

We have the following upper bound for the value of the loss function at the initial point.

**Lemma E.5.** *Initialize $W_0$ by (5). Let $t_5 > 0$. We have*

$$L_n(W_0) \leq \frac{2\sigma_3^2 B_\sigma^2}{C} \ln \frac{2}{t_5} + 2B_Y^2$$

*with probability at least $1 - t_5$ over $W_0$.*

*Proof.*

$$L_n(W_0) = \frac{1}{n} \sum_{i=1}^n [f_{W_0}(X_i) - Y_i]^2 \leq \frac{2}{n} \sum_{i=1}^n |f_{W_0}(X_i)|^2 + 2B_Y^2.$$

By Proposition 2.6.1 in Vershynin (2018),

$$\|f_{W_0}\|_{\psi_2}^2 = \left\| \frac{1}{\sqrt{m_2}} \sum_{k_2=1}^{m_2} (w_0)_{k_2}^{(3)} f_{k_2}^{(2)} \right\|_{\psi_2}^2 \leq \frac{C}{m_2} \sum_{k_2=1}^{m_2} \left| f_{k_2}^{(2)} \right|^2 \left\| (w_0)_{k_2}^{(3)} \right\|_{\psi_2}^2 \leq C\sigma_3^2 B_\sigma^2.$$

Hence by Proposition 2.5.2 in Vershynin (2018), we have $|f_{W_0}| \leq \tau$ with probability at least $1 - 2e^{\frac{-C\tau^2}{\sigma_3^2 B_\sigma^2}}$ over $W_0$, which implies the lemma. $\square$

The following lemma is a useful result for proving global convergence in optimization theory.

**Lemma E.6.** *Let $n \in \mathbb{N}_+$. Given $x, y \in \mathbb{R}^n$. For a differentiable function $f : \mathbb{R}^n \to \mathbb{R}$, if there exists some constant $L$ such that for any $t \in [0, 1]$,*

$$\|\nabla f[tx + (1-t)y] - \nabla f(x)\|_2 \leq L\|y - x\|_2,$$

*then*

$$f(y) \leq f(x) + \nabla f(x)^T (y - x) + \frac{L}{2} \|y - x\|_2^2.$$

With the above preparations, we can now prove Theorem 4.

*Proof of Theorem 4.* From Lemma E.5 we know

$$|L_n(W_0)| \leq \frac{2\sigma_3^2 B_\sigma^2}{C} \ln \frac{2}{t_5} + 2B_Y^2 \tag{E.13}$$

with probability at least $1 - t_5$ over $W_0$. By the update rule of gradient decent, Lemma E.2, (E.5), (E.13), we have with probability at least $1 - t_2 - t_3 - t_5$ over $W_0$,

$$|w_1 - w_0| \leq \eta \left| \frac{\partial L_n(W_0)}{\partial w} \right| \leq \frac{2\eta \widetilde{m}_2 B_\sigma B_{\sigma'}^2 B_x B_3(W_0) B_2(W_0) L_n^{1/2}(W_0)}{\sqrt{m_2}}$$

$$\leq \frac{2\eta \widetilde{m}_2 B_\sigma B_{\sigma'}^2 B_x B_3^{(t_2)} B_2^{(t_3)} \left( \frac{2\sigma_3^2 B_\sigma^2}{C} \ln \frac{2}{t_5} + 2B_Y^2 \right)^{1/2}}{\sqrt{m_2}} \leq \min \left\{ B_1^{(t_6)}, B_2^{(t_3)}, B_3^{(t_2)} \right\}$$

provided

$$\sqrt{m_2} \geq \frac{2\eta \widetilde{m}_2 B_\sigma B_{\sigma'}^2 B_x B_3^{(t_2)} B_2^{(t_3)} \left( \frac{2\sigma_3^2 B_\sigma^2}{C} \ln \frac{2}{t_5} + 2B_Y^2 \right)^{1/2}}{\min \left\{ B_1^{(t_6)}, B_2^{(t_3)}, B_3^{(t_2)} \right\}}.$$

Hence with probability at least $1 - t_2 - t_3 - t_5 - t_6$ over $W_0$,

$$B_3(W_1) = \max \left\{ \max_{k_2} \left| (w_1)_{k_2}^{(3)} \right|, 1 \right\} \leq \max \left\{ \max_{k_2} \left| (w_1)_{k_2}^{(3)} - (w_0)_{k_2}^{(3)} \right| + \left| (w_0)_{k_2}^{(3)} \right|, 1 \right\}$$

$$\leq \max \left\{ 2B_3^{(t_2)}, 1 \right\} = 2B_3^{(t_2)} \tag{E.14}$$

and similarly $B_2(W_1) \le 2B_2^{(t_3)}, B_1(W_1) \le 2B_1^{(t_6)}$. Setting $y = W_1, x = W_0$ in Lemma E.6 and applying Lemma E.3, we have

$$L_n(W_1) \le L_n(W_0) + \nabla L_n(W_0)^T(W_1 - W_0) + \frac{L_0}{2}\|W_1 - W_0\|_2^2$$

$$\le L_n(W_0) + \nabla L_n(W_0)^T(W_1 - W_0) + \frac{L}{2}\|W_1 - W_0\|_2^2$$

$$= L_n(W_0) - \left(\eta - \frac{\eta^2 L}{2}\right)\|\nabla L_n(W_0)\|_2^2,$$

with probability at least $1 - t_2 - t_3 - t_5 - t_6$ over $W_0$, where in the second step we use the fact that

$$L_0 = C_4\left[\sqrt{\frac{86}{m_2}}\widetilde{m}_2\widetilde{m}_1 m_0 L_n^{1/2}(W_0) + \sqrt{5\widetilde{m}_2\widetilde{m}_1 m_0(2\widetilde{m}_1 + \widetilde{m}_1 m_0 + 2)}\right]B_3(W_0)B_3(W_1)\max\left\{B_2^2(W_0), B_2^2(W_1)\right\}$$

$$\le 16C_4\left[\sqrt{\frac{86}{m_2}}\widetilde{m}_2\widetilde{m}_1 m_0\left(\frac{2\sigma_3^2 B_\sigma^2}{C}\ln\frac{2}{t_5} + 2B_Y^2\right)^{1/2} + \sqrt{5\widetilde{m}_2\widetilde{m}_1 m_0(2\widetilde{m}_1 + \widetilde{m}_1 m_0 + 2)}\right]\left(B_3^{(t_2)}\right)^2\left(B_2^{(t_3)}\right)^2 = L.$$

and the third step is due to the update rule of gradient descent. Note that our choice of $m$ satisfies the condition in Lemma E.4. So applying Lemma E.4 to the term $\|\nabla L_n(W_0)\|_2^2$ and using union bound, we obtain with probability at least $1 - t_2 - t_3 - t_4 - t_5 - t_6$ over $W_0$ that

$$L_n(W_1) \le \left(1 - \eta\mu + \frac{1}{2}\eta^2\mu L\right)L_n(W_0). \tag{E.15}$$

Now let's consider next iteration. With probability at least $1 - t_2 - t_3 - t_4 - t_5 - t_6$ over $W_0$,

$$|w_2 - w_0| \le \eta\left|\frac{\partial L_n(W_1)}{\partial w}\right| + \eta\left|\frac{\partial L_n(W_0)}{\partial w}\right|$$

$$\le \frac{2\eta\widetilde{m}_2 B_\sigma B_{\sigma'}^2 B_x B_3(W_1)B_2(W_1)L_n^{1/2}(W_1)}{\sqrt{m_2}} + \frac{2\eta\widetilde{m}_2 B_\sigma B_{\sigma'}^2 B_x B_3(W_0)B_2(W_0)L_n^{1/2}(W_0)}{\sqrt{m_2}}$$

$$\le \frac{8\eta\widetilde{m}_2 B_\sigma B_{\sigma'}^2 B_x B_3(W_0)B_2(W_0)}{\sqrt{m_2}}\left[\left(1 - \eta\mu + \frac{1}{2}\eta^2\mu L\right)^{1/2} + 1\right]L_n^{1/2}(W_0)$$

$$\le \frac{16\widetilde{m}_2 B_\sigma B_{\sigma'}^2 B_x B_3^{(t_2)}B_2^{(t_3)}\left(\frac{2\sigma_3^2 B_\sigma^2}{C}\ln\frac{2}{t_5} + 2B_Y^2\right)^{1/2}}{\sqrt{m_2}(\mu - \frac{1}{2}\eta\mu L)} \le \min\left\{B_1^{(t_6)}, B_2^{(t_3)}, B_3^{(t_2)}\right\},$$

where we use Lemma E.2, (E.14)-(E.15), (E.13) and the relation

$$\sqrt{m_2} \ge \frac{16\widetilde{m}_2 B_\sigma B_{\sigma'}^2 B_x B_3^{(t_2)}B_2^{(t_3)}\left(\frac{2\sigma_3^2 B_\sigma^2}{C}\ln\frac{2}{t_5} + 2B_Y^2\right)^{1/2}}{(\mu - \frac{1}{2}\eta\mu L)\min\left\{B_1^{(t_6)}, B_2^{(t_3)}, B_3^{(t_2)}\right\}}$$

in the second, third, fourth and last step, respectively.

Assume that with probability at least $1 - t_2 - t_3 - t_4 - t_5 - t_6$ over $W_0$,

$$|w_\tau - w_0| \le \min\left\{B_1^{(t_6)}, B_2^{(t_3)}, B_3^{(t_2)}\right\}, \quad \tau = 1, 2, \cdots, t, \tag{E.16}$$

$$L_n(W_\tau) \le \left(1 - \eta\mu + \frac{1}{2}\eta^2\mu L\right)L_n(W_{\tau-1}), \quad \tau = 1, 2, \cdots, t-1 \tag{E.17}$$

and we prove the case for $|w_{t+1} - w_0|$ and $L_n(W_t)$. From (E.16) we immediately obtain with probability at least $1 - t_2 - t_3 - t_4 - t_5 - t_6$ over $W_0$,

$$B_3(W_\tau) = \max\left\{\max_{k_2}\left|(w_\tau)_{k_2}^{(3)}\right|, 1\right\} \le \max\left\{\max_{k_2}\left|(w_\tau)_{k_2}^{(3)} - (w_0)_{k_2}^{(3)}\right| + \left|(w_0)_{k_2}^{(3)}\right|, 1\right\}$$

$$\le \max\{2B_3^{(t_2)}, 1\} = 2B_3^{(t_2)} \tag{E.18}$$

and similarly $B_2(W_\tau) \le 2B_2^{(t_3)}, B_1(W_\tau) \le 2B_1^{(t_6)}$ for $\tau = 0, 1, \cdots, t$. Setting $y = W_t, x = W_{t-1}$ in Lemma E.6 and applying Lemma E.3, we have with probability at least $1 - t_2 - t_3 - t_4 - t_5 - t_6$ over $W_0$,

$$L_n(W_t) \le L_n(W_{t-1}) + \nabla L_n(W_{t-1})^T(W_t - W_{t-1}) + \frac{L_{t-1}}{2}\|W_t - W_{t-1}\|_2^2$$

$$\le L_n(W_{t-1}) + \nabla L_n(W_{t-1})^T(W_t - W_{t-1}) + \frac{L}{2}\|W_t - W_{t-1}\|_2^2$$

$$= L_n(W_{t-1}) - \left(\eta - \frac{\eta^2 L}{2}\right)\|\nabla L_n(W_{t-1})\|_2^2,$$

where in the second step we use the fact that

$$L_{t-1} =$$

$$C_4\left[\sqrt{\frac{86}{m_2}}\widetilde{m}_2\widetilde{m}_1 m_0 L_n^{1/2}(W_{t-1}) + \sqrt{5\widetilde{m}_2\widetilde{m}_1 m_0(2\widetilde{m}_1 + \widetilde{m}_1 m_0 + 2)}\right] B_3(W_{t-1})B_3(W_t)\max\left\{B_2^2(W_{t-1}), B_2^2(W_t)\right\}$$

$$\le 16C_4\left[\sqrt{\frac{86}{m_2}}\widetilde{m}_2\widetilde{m}_1 m_0\left(\frac{2\sigma_3^2 B_\sigma^2}{C}\ln\frac{2}{t_5} + 2B_Y^2\right)^{1/2} + \sqrt{5\widetilde{m}_2\widetilde{m}_1 m_0(2\widetilde{m}_1 + \widetilde{m}_1 m_0 + 2)}\right]\left(B_3^{(t_2)}\right)^2\left(B_2^{(t_3)}\right)^2 = L.$$

The assumption that $|w_{t-1} - w_0| \le \min\left\{B_1^{(t_6)}, B_2^{(t_3)}, B_3^{(t_2)}\right\}$ enables us to apply Lemma E.4 to the term $\|\nabla L_n(W_{t-1})\|_2^2$. Thus we obtain

$$L_n(W_t) \le \left(1 - \eta\mu + \frac{1}{2}\eta^2\mu L\right)L_n(W_{t-1}) \tag{E.19}$$

with probability at least $1 - t_2 - t_3 - t_4 - t_5 - t_6$ over $W_0$. For the distance between $W_{t+1}$ and $W_0$, with the same probability,

$$|w_{t+1} - w_0| \le \eta\sum_{\tau=0}^t\left|\frac{\partial L_n(W_\tau)}{\partial w}\right| \le \frac{2\eta\widetilde{m}_2 B_\sigma B_{\sigma'}^2 B_x}{\sqrt{m_2}}\sum_{\tau=0}^t B_3(W_\tau)B_2(W_\tau)L_n^{1/2}(W_\tau)$$

$$\le \frac{8\eta\widetilde{m}_2 B_\sigma B_{\sigma'}^2 B_x B_3(W_0)B_2(W_0)L_n^{1/2}(W_0)}{\sqrt{m_2}}\sum_{\tau=0}^t\left(1 - \eta\mu + \frac{1}{2}\eta^2\mu L\right)^{\tau/2}$$

$$\le \frac{8\eta\widetilde{m}_2 B_\sigma B_{\sigma'}^2 B_x B_3(W_0)B_2(W_0)\left(\frac{2\sigma_3^2 B_\sigma^2}{C}\ln\frac{2}{t_5} + 2B_Y^2\right)^{1/2}}{\sqrt{m_2}}\frac{1}{1 - \left(1 - \eta\mu + \frac{1}{2}\eta^2\mu L\right)^{1/2}}$$

$$\le \frac{16\widetilde{m}_2 B_\sigma B_{\sigma'}^2 B_x B_3^{(t_2)}B_2^{(t_3)}\left(\frac{2\sigma_3^2 B_\sigma^2}{C}\ln\frac{2}{t_5} + 2B_Y^2\right)^{1/2}}{\sqrt{m_2}(\mu - \frac{1}{2}\eta\mu L)}$$

$$\le \min\left\{B_1^{(t_6)}, B_2^{(t_3)}, B_3^{(t_2)}\right\},$$

where we use Lemma E.2, (E.17)-(E.19), (E.13) and the relation

$$\sqrt{m_2} \ge \frac{16\widetilde{m}_2 B_\sigma B_{\sigma'}^2 B_x B_3^{(t_2)}B_2^{(t_3)}\left(\frac{2\sigma_3^2 B_\sigma^2}{C}\ln\frac{2}{t_5} + 2B_Y^2\right)^{1/2}}{(\mu - \frac{1}{2}\eta\mu L)\min\left\{B_1^{(t_6)}, B_2^{(t_3)}, B_3^{(t_2)}\right\}}$$

in the second, third, fourth and last step, respectively. Hence we complete the proof.

$$\square$$