# OpenReview forum: "Estimation error of gradient descent in deep regressions"
_ICLR.cc/2024/Conference — Submitted to ICLR 2024_

### Official Review · Reviewer_x5tH · 2023-10-17

**Soundness:** 3 good
**Presentation:** 2 fair
**Contribution:** 3 good
**Rating:** 6
**Confidence:** 4

**Summary:**

This paper gives an upper bound of the estimation error of a three-layer sigmoidal neural network (NN) with GD training on regression task. The estimation error is defined to be $\|\hat{f}\_t-f_*\|^2$ where $f_*$ is the target function and $\hat{f}_t$ is the empirical regressor after t GD steps, which can be further upper bounded the approximation error, plus the generalisation error and plus the optimisation error.

The paper devotes itself into upper bounding the above three errors by various techniques:
- approximation error: Approximation of Lipschitz function
- generalisation error: symmetrisation and Rademacher Complexity
- optimisation error: LP-type inequality

The final result (Theorem) is the combination of the above bounds.

**Strengths:**

Originality: This paper takes an ambitious approach to tackle the approximation error, generalisation error and the optimisation error all at once. Combining previous results from different areas in a concrete NN setting is surely novel.

Quality: This paper applies a plenty of technical lemmata to provide a rigorous proof. As said, the bound in this paper is still far from the minimax optimal rate, but it is a good attempt.

Clarity: All the constants are tracked, and definitions are given in the main text or in the appendix, thus the paper is self-containing.

Significance: The field of machine learning theory has devoted much effort on approximation error, generalisation error and the optimisation error separately. It is encouraging to combine previous work to give a bound on the estimation (total) error. Also, the main theorem gives a good scaling on the hyper parameters like width, initialisation, learning rate for 3-layer sigmoidal NN, which could serve as theoretical and empirical reference for future research.

**Weaknesses:**

A few minor weaknesses hinders the clarity of this paper.

1. The title is misleading, in my point of view. Better state as "Estimation Error of Gradient Descent in 3-layer sigmoidal neural network". I would expect the result could extend to multiple layer L of NN when one states "deep" regression. Also, since this paper deals with 3-layer NN. I wonder if there is already similar result for 2-layer NN, which I cannot find this paper citing.

2. I understand there are many constants and symbols in the paper due to the circumstances of the paper, but I find it confusing to have $L,L_n,L_\sigma$ all representing different things.

3. There is no experimental result supporting the claim.

**Questions:**

1. What is L in Theorem 1?

2. Is there already similar result for 2-layer NN? Or can we adjust the result of this paper to obtain that for 2-layer NN? What would be the difference?

3. Related to above question, can one extend the proof in this paper to any deep NN?

4. Optimisation error is proved using PL condition that the optimal weight is near to the initialisation. Is it too restrictive for realistic case?

5. Could you provide any experimental result? Even if the bound is very loose, it could give a hint on where one could tighten the bound, on approximation, generalisation or optimisation.

---

### Official Review · Reviewer_Mz9q · 2023-10-22

**Soundness:** 2 fair
**Presentation:** 3 good
**Contribution:** 2 fair
**Rating:** 3
**Confidence:** 3

**Summary:**

This paper analyzes a fully connected ReLU activated neural network of depth 3, with a few reasonable assumptions about the width and scale of parameters. With parameterization which means the hidden layers have $m_1, m_2 > n,d$ dimensions, the true relation between feature and label can be learned with small error.

**Strengths:**

This paper gives concrete proof based on error decomposition and analyzing each term. It is correct and novel. It is comprehensive that covers different aspects, like each term in error decomposition, statistical error and algorithm analysis, etc. The proof in appendix is structured clearly.

**Weaknesses:**

The proof is based on neural tangent kernel theory which is used in a few NN theory papers, and the proof is similar to the idea that the local landscape of the objective function is almost linear and the optimal points are dense when the NN is overparameterized. Although the paper is thorough and comprehensive, I’m not sure how much novelty it provides compared to previous work technically, and whether the bound, for example, the lower bound of the widths of each layer, is optimal compared with previous papers.

Is the novelty about constraints? Does the optimization path interact with the boundary of feasible set, or the feasible set is so large that once the initial point is far from boundary, since it moves a small distance, the constraints are never hit?

The NN structure is quite special that each layer has to be some sort of shape and the architecture is simple. As practical NN is developing, it would make more sense to present a generic method or discussion that can be applied to many types of NNs.

It is fine that the bound terms are complicated and not intuitive, but it makes the paper readable if we see more discussions, like which term is bottleneck, how to insert some intermediate values (such as $B$ with different sub/superscripts) to get the bound in informal Thm.

It is better to give lower bound of $n$ rather saying “sufficiently large”.

I am not sure if any practical NN applications have "iterates moving locally" so that one can use kernels, and people usually use regularizers. Is there an experiment or practical observation that verifies the phenomenon in this paper in practice?

**Questions:**

When discussing the generalization error, do we use the empirical loss of the $n$ training samples when defining $L_n$, or fresh new samples? Since the optimizer depends on the training samples, I guess one shall define on test set which is independent. However I believe the error should be well bounded in both cases.

---

### Official Review · Reviewer_JNjG · 2023-10-24

**Soundness:** 3 good
**Presentation:** 3 good
**Contribution:** 2 fair
**Rating:** 5
**Confidence:** 3

**Summary:**

This paper theoretically analyzes the approximation, optimization and generalization error of a three-layer neural network model class.
Specifically, this paper firstly shows the universal approximation property of a three-layer neural network, then proves that gradient descent enjoys a linear convergence rate in the overparameterization regime, and finally gives a generalization error bound based on the parameter norm during training.

**Strengths:**

* The paper is written clearly and the ideas have been presented in the proper order.
* The combined analysis of approximation, optimization and generalization is important as it offers an improved insight into the interaction among these three errors.

**Weaknesses:**

* The detailed comparison with previous results is missing. For example, for the optimization result, this paper proves a result that in the overparameterization regime, the loss function satisfies the PL condition along the gradient descent dynamics. This result is also stated in [1], so how does the current result differ with [1]?
Also, for the generalization result, there are some generalization bounds for three-layer neural networks  (or more general deep neural networks), e.g., [2,3,4]. What is the difference between this generalization result to these previous bounds? From my understanding, the generalization bound in this paper is based on the distance $|w_t-w_0|$, and there are many existing results for distance-based generalization bounds, e.g., [5]. What is the difference for the techniques on the distance-based generalization bound between the two works?


* For the generalization result, I think it would be more convincing if the authors could provide numerical experiments on the tightness of the generalization bound.






[1] Chaoyue Liu, Libin Zhu, and Mikhail Belkin. Loss landscapes and optimization in over-parameterized non-linear systems and neural networks. *Applied and Computational Harmonic Analysis*, 59: 85–116, 2022.

[2] Gatmiry, Khashayar, Stefanie Jegelka, and Jonathan Kelner. "Adaptive Generalization and Optimization of Three-Layer Neural Networks." *The Tenth International Conference on Learning Representations (ICLR)*. 2022.

[3] Ju, Peizhong, Xiaojun Lin, and Ness Shroff. "On the generalization power of the overfitted three-layer neural tangent kernel model." *Advances in Neural Information Processing Systems 35* (2022): 26135-26146.

[4] Wang, Puyu, et al. "Generalization Guarantees of Gradient Descent for Multi-Layer Neural Networks." *arXiv preprint arXiv:2305.16891* (2023).

[5] Arora, Sanjeev, et al. "Fine-grained analysis of optimization and generalization for overparameterized two-layer neural networks." *International Conference on Machine Learning. PML*R, 2019.

**Questions:**

I have one question on the setting of the initialization scheme. From equation (5), it seems that the variance of the initialization is set to be constant with respect to $d$ and $m_2$.
But in the practical training, the common initialization scheme follows from the Kaiming initialization [6] or the Xavier initialization [7] to keep the variance perseverance, i.e., the variance of the initialization scales like $\mathcal{\Theta}(1/m)$ for width $m$.
So I am wondering why this paper adopts the constant variance scheme for training, and why does this constant variance scheme guarantee the convergence of gradient descent?




[6] He, Kaiming, et al. "Delving deep into rectifiers: Surpassing human-level performance on imagenet classification." *Proceedings of the IEEE international conference on computer vision*. 2015.

[7] Glorot, Xavier, and Yoshua Bengio. "Understanding the difficulty of training deep feedforward neural networks." *Proceedings of the thirteenth international conference on artificial intelligence and statistics. JMLR Workshop and Conference Proceedings*, 2010.

---

### Official Review · Reviewer_VTNi · 2023-10-31

**Soundness:** 3 good
**Presentation:** 1 poor
**Contribution:** 2 fair
**Rating:** 3
**Confidence:** 3

**Summary:**

This work simultaneously studies approximation, optimization and generalization error of GD applied to three-layer neural networks.

**Strengths:**

1. This work bridges the gap between previous works and the goal of studying the approximation, optimization and generalization error as a whole.

**Weaknesses:**

1. The main Theorem (either Theorem 1 or Theorem 5) requires much more discussion. At least two aspects of discussion are missing: (i) what the results indicate, and (ii) how the results are compared to previous results quantitatively.
2. There are a lot of notations in Section 3.2, and I would be happy to see more explanations about what how these notations mean. For example, $L$ looks like a Lipschitz constant and $\mu$ looks like the coefficient of strong convexity. If the use of these notations are too technical, I would recommend putting them in the appendices, and the author could just mention the existence of these requirements in the main text.
3. The notations of $I_{k_1}^{(1)}$ $I_{k_2}^{(2)}$ are a bit confusing to me. I failed to find $I_{k_1}^{(1)}$ used anywhere in the main text. Also, the definition of the neural networks in Eq. (2) and how $I_{k_2}^{(2)}$ is used look a bit strange to me
4. The comments after Theorem 3 claims that Theorem 3 is a result of **constrained** optimization. However, boundedness is hardly enforced in practice, and I am skeptical about whether this is a contribution.
5. The motivation of the authors studying a **3-layer** network is a bit unclear to me. I am expecting to see the technical contributions if the analysis of three-layer networks embodies some intrinsic difficulties (and the proof sketch should probably highlight the technical contributions). I am also curious about the method in this work can be extended to networks of arbitrary depth.

**Questions:**

1. Could the authors provide more details about how to obtain Theorem 1 from Theorem 5? It does not seem quite obvious to me.
2. As the indications of the main result appear a bit unclear to me, does the result obtained in this work indicate that there is extra tradeoff between optimization, approximation and generalization compared to the scenario when we only consider two of the aspects?

---

### Meta-Review · Area_Chair_zhRS · 2023-12-11

**Metareview:**

3 reviewers out of 4 recommended a rejection. The other reviewer was not strongly arguing for acceptance. There was no rebuttal answering the reviewers questions. The AC agrees with the majority and recommends rejection. Concerns were expressed about novelty, presentation and missing analysis. The authors are encouraged to take the reviewers comments in consideration for a major revision.

**Justification For Why Not Higher Score:**

Paper is not ready for ICLR.

**Justification For Why Not Lower Score:**

N/A

---

### Decision · Program_Chairs · 2024-01-16

Reject